# Strain-gradient mediated local conduction in strained bismuth ferrite films

Ming-Min Yang[1], Affan N. Iqbal [1], Jonathan J.P. Peters[1], Ana M. Sanchez [1] & Marin Alexe[1]

It has been recently shown that the strain gradient is able to separate the light-excited electron-hole pairs in semiconductors, but how it affects the photoelectric properties of the photo-active materials remains an open question. Here, we demonstrate the critical role of the strain gradient in mediating local photoelectric properties in the strained $BiFeO_3$ thin films by systematically characterizing the local conduction with nanometre lateral resolution in both dark and illuminated conditions. Due to the giant strain gradient manifested at the morphotropic phase boundaries, the associated flexo-photovoltaic effect induces on one side an enhanced photoconduction in the $R$-phase, and on the other side a negative photo-conductivity in the morphotropic $T'$-phase. This work offers insight and implication of the strain gradient on the electronic properties in both optoelectronic and photovoltaic devices.

---

[1] Department of Physics, The University of Warwick, Coventry CV4 7AL, UK. Correspondence and requests for materials should be addressed to M.-M.Y. (email: Mingmin.Yang.1@warwick.ac.uk) or to M.A. (email: M.Alexe@warwick.ac.uk)

Exploring and understanding the correlations between structures and physical properties in condensed materials have been a long-term fundamental topic in modern science. Among all the potential stimuli, strain, especially in the epitaxial oxide thin films, has been demonstrated as an effective tool in modulating the crystallographic structures as well as physical properties, with many notable examples including inducing ferroelectricity in paraelectric materials[1], stabilization of super-tetragonal structures in ferroelectric films[2,3] and doubling the critical temperature in superconductors[4]. In contrast, the strain-gradient, i.e. the derivative of the strain, has been largely overlooked in the past despite its ubiquitous existence in film materials. This is mainly due to the arbitrary assumption of its negligible effect. However, with the miniaturization of device sizes, the magnitude of strain-gradient along with its associated effects turns out to be significant when reducing material dimension into nanometre range. For example, the strain gradient in the out-of-plane direction of perovskite oxide thin films can reach $10^6 m^{-1}$ due to in-plane strain relaxation and its value at the dislocation core can even reach $10^9 m^{-1}$ [5,6]. Due to the flexoelectric effect, those strain gradients would alter domain structures or modify hysteresis curves of ferroelectric thin films and even induce a large electric polarization in paraelectric crystals[5–10]. Apart from this electromechanical coupling, the strain gradient can also affect electronic properties via redistributing charged ionic defects, such as oxygen vacancies in oxide materials[11–14].

We have recently demonstrated that the strain gradient is also able to separate photo-excited electron-hole pairs, similar to chemical gradients and electric fields, giving rise to a unique photovoltaic effect termed flexo-photovoltaic effect[15]. The flexo-photovoltaic effect is allowed by symmetry in all materials. In the case of epitaxial/polycrystalline solar cells and light sensors, crystallographic disorders (such as dislocations and grain boundaries) and the lattice mismatch at the interfaces generate substantial strain gradients. The associated flexo-photovoltaic effect would have a considerable impact on the electronic properties and performance of these devices. However, this topic remains largely unexplored.

Concerning the studies of the strain and strain gradient associated effects, the $BiFeO_3$ thin films grown on (001)-oriented $LaAlO_3$ substrates offer a fertile playground. Due to the large lattice mismatch between $BiFeO_3$ and $LaAlO_3$ substrate, the large in-plane compressive strain imposed on the $BiFeO_3$ film enables the stabilization of a tetragonal-like monoclinic phase (termed for simplicity $T$-matrix phase) with an enhanced $c/a$ ratio of 1.23[2,16]. With increasing film thickness, another structure variant, namely

the needle-shaped rhombohedral-like phase (termed $R$-phase), appears in the $T$-matrix phase which is epitaxially confined between tilted $T$-phases (termed $T'$-phase). Due to the large lattice mismatch and epitaxial confinement, a giant strain as well as strain gradient is generated at the interface between these two morphotropic phases. Additionally, by virtue of its moderate bandgap, $BiFeO_3$ thin films can absorb visible light and generate photoconductive effects as well as more intriguing photoelectric effects, such as persistent conductivity[17]. Thus, these morphotropic phases in strained $BiFeO_3$ films with the same chemical composition provide an excellent platform to study the effects of strain gradient, especially the associated flexo-photovoltaic effect, on photoelectric properties in oxide materials. Prior work observed an enhancement of the photocurrent at the regions where $T'/R$ mixed phases coexist in a capacitor geometry characterized by a confocal microscopy[18]. However, the sub-micrometre resolution of the confocal microscopy can hardly unravel detailed electronic features of the nanoscale morphotropic entities with a lateral dimension less than one hundred nanometres. On the other hand, the thickness-resolved study on the strained $BiFeO_3$ films with coplanar electrodes reveals a decreasing trend of the photoconduction with increasing density of the morphotropic phase boundaries[19]. Therefore, the real role played by the morphotropic phase boundaries in these devices remains yet to be resolved. In this work, by directly characterizing the local electronic properties of these morphotropic phases with a nanometre resolution, we shed light on the critical importance of the strain gradient in mediating local photoelectric properties and elucidate the exact role of the morphotropic phase boundaries.

## Results

**Characterization of the dark conduction.** We first characterized the electrical conduction on a $BiFeO_3$ thin films by a conventional conductive atomic force microscopy (cAFM) performed without any illumination (in dark conditions). During the measurement, the surface topography and the local drift current are simultaneously mapped, enabling us to identify the electronic conduction of these nanoscale entities. As shown in Fig. 1, the cAFM tip probes certain current over the whole scanned area, including $T$-matrix phase and $R/T'$ mixed phase regions. The dark current probed in $R/T'$ mixed region exhibits needle-like features similar to that of the surface topography. To reveal detailed information regarding the respective dark conduction of $R$- and $T'$-phases, we compared the line profiles of the dark current with its corresponding topography (marked by the blue arrow in Fig. 1a). As

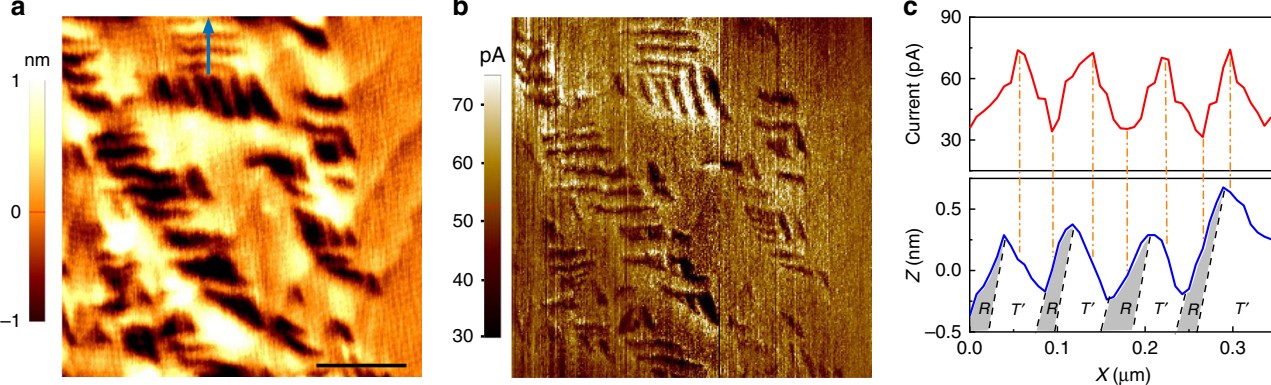

**Fig. 1** Spatially resolved dark current distribution. **a** Surface topography and **b** corresponding dark current distribution mapped on the $BiFeO_3$ (60 nm)/ $La_{0.7}Sr_{0.3}MnO_3$ (5 nm)/$LaAlO_3$ film without any illumination. Scale bar 500 nm. **c** Dark current and surface morphology profile comparison of the area marked by blue arrow in (**a**). The current is acquired by applying 2 V to the bottom electrode with the conductive tip virtually grounded

illustrated in Fig. 1c, the dark current reaches maxima in the middle of $T'$-phases while decreases to minima in the middle of $R$-phases. Previous cAFM studies of the strained $BiFeO_3$ thin films hypothesised an enhanced conduction at one type of the morphotropic phase boundaries in comparison to other entities including $T$-matrix, $R$- and $T'$-phases[20]. However, this model cannot explain the dark current distribution mapped in this work due to two reasons. Firstly, if the dark current variation is due to the enhanced conduction at morphotropic phase boundaries, the maxima of the dark current should correspond to the positions where the phase boundaries are located as is the case of ferro-electric domain walls[21]. However, the dark current maxima mapped here appear at the middle of the $T'$-phase rather than at the phase boundaries (see Supplementary Note 1). Secondly, further cAFM scans reveal that the dark current probed on pure $T$-phase matrix, where no phase boundaries exist, is between the current detected on the $R$- and $T'$-phase in the mixed phase region (See Supplementary Fig. 2). Instead, these results strongly argue that the $R$-phase matrix has a lower conductivity (lower current) than the $T$-matrix phase, while the tilted $T'$-phase has a higher conductivity than the $T$-matrix phase.

**Spatially resolved distribution of the photoconduction.** To characterize the photoelectric properties of the morphotropic phases, we mapped the distribution of the drift current under illumination (i.e. photocurrent) by using the photoelectric atomic force microscopy (Ph-AFM). The Ph-AFM consists of an AFM-based system modified by a custom current amplifier/filter system and an optical system. The latter allows illumination of the sample surface with tuneable light polarizations by employing a half-wavelength ($\lambda/2$) plate[22–24]. Details of the Ph-AFM system is given in the Methods section. The $BiFeO_3$ surface is illuminated by a blue laser with a wavelength of 405 nm ($h\nu = 3.06$ eV) and an intensity of 1 W cm$^{-2}$. As demonstrated by the $I$-$V$ characteristics (Supplementary Fig. 3) acquired under illumination, the local photocurrent probed by the AFM tip increases almost linearly with applied voltage, indicating an Ohmic-like behaviour of the electronic transport. This enables us to study the intrinsic photoelectric properties of this oxide system. During the photo-current mapping by the Ph-AFM system, a bias of 2 V was applied to the in-plane side electrode (see "Methods") which is well below the threshold voltage to induce the phase transition between $T$- and $R$-phases[25]. Note that the conductive tip probes negligible current without applying external bias under illumi-nation (See Supplementary Fig. 4). Figure 2a shows the surface topography of a scanned area and its corresponding spatially resolved photocurrent distribution is shown in Fig. 2b. Clearly, similar to the surface morphology, the spatially resolved photo-current distribution also shows stripe-like patterns which exhibits some intriguing features. Firstly, the photocurrent is detected over the whole scanned area, indicating the photo-sensitive nature of the strained $BiFeO_3$ irrespective of different lattice structures. Secondly, the bright stripes in Fig. 2b indicate that the photo-current is significantly enhanced at some areas of the $R/T'$- mixed phase regions. Additionally, dark contrast stripes are shown to be located between the bright stripes, indicating suppressed photo-current also happens in the mixed phase regions. In order to precisely correlate the photocurrent variation with morphotropic structures, we analyzed a profile scan of both topography and photocurrent of an area marked by the blue arrow in Fig. 2a. As shown in Fig. 2c, the left part of the graph corresponds to the $R$-/$T'$ mixed phase region while the right part is the pure $T$-matrix phases. When scanning the tip from the $T$-matrix phase to the mixed phase region, the conductive AFM tip first probes a moderate photocurrent of ~20 pA on the $T$-matrix phases while the photocurrent increases sharply when it approaches the $R$-phase, reaching a maximum in the middle of the $R$-phase. Then the photocurrent decreases on the tilted $T'$-phase and then increases again in the $R$-phase in an alternating fashion. The minimum photocurrent, probed on $T'$-phase, is clearly smaller than that probed on the $T$-matrix phase. It is worthwhile noting that both enhancement and suppression of the photocurrent at the mixed phase region are not only confined to the areas close to the phase boundaries but also extends to the whole volume of morphotropic phases. Therefore, the Ph-AFM scans demonstrate that, in respect to the $T$-matrix phase, the mor-photropic $R$-phase shows enhanced photoconduction while the tilted $T'$-phase confined by the $R$-phase exhibits a largely supressed photoconduction.

Keeping in perspective the enhanced dark conduction of the $T'$-phase compared to that of the $R$-phase, the reversal of the conduction contrast under illumination revealed above is clearly abnormal. In the wide bandgap oxide materials, light-induced band-band transition would dramatically increase the none-quilibrium carrier density and enhance the conduction. Usually in these oxides a higher dark conductivity is generally associated with a higher photoconductivity. For example, the ferroelectric domain walls in perovskite oxides possess both enhanced dark conduction and photoconduction in comparison to the domain bulk[21,22]. This abnormal photoelectric behaviour of the titled $T'$-phase is similar to the negative photoconductivity observed in

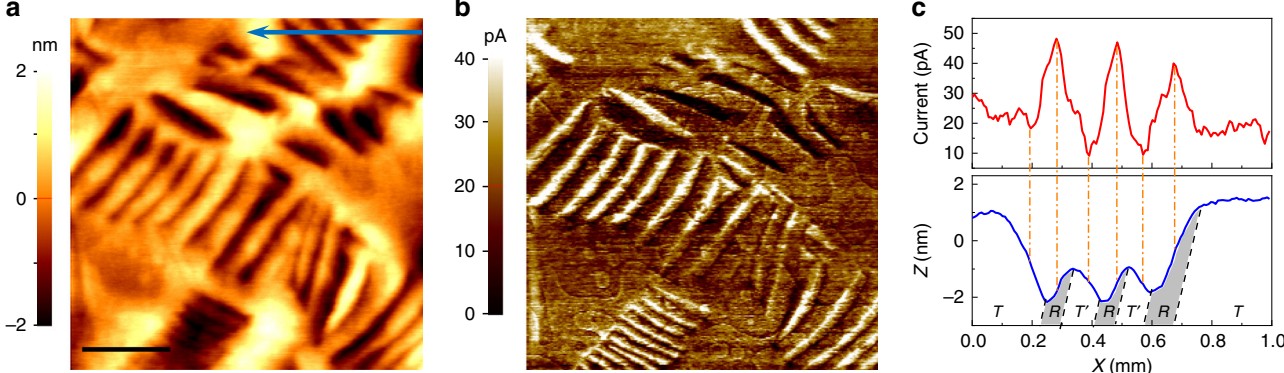

**Fig. 2** Spatially resolved photocurrent distribution. **a** Surface topography and **b** photocurrent distribution characterized under illumination on a 100 nm-thick $BiFeO_3$/$LaAlO_3$ thin film; Scale bar 500 nm. **c** Profile comparison between the photocurrent and surface morphology of the area marked by blue arrow in (a). The photocurrent is acquired under the illumination of 405 nm light with an intensity of 1 W cm$^{-2}$. The bias is applied to a side Pt electrode evaporated on the surface of the $BiFeO_3$ film with the conductive tip virtually grounded

Ge and Si with deep impurity levels, wherein illumination with infrared radiation decreases the conduction[26,27]. In the cases of Ge and Si, minority carriers are optically freed by light with photon energy well below the semiconductor bandgap, which enhances the recombination process with the majority carriers and thus reduces the conduction[28]. It is worthwhile noting the 405 nm laser used in this work enables the band-band excitation of non-equilibrium carriers in the all the morphotropic phases, which however would not induce the negative photo-conductivity effect by itself. More intriguingly, despite sharing the same chemical composition and crystallographic structure, the $T$-matrix phase and tilted $T'$-phases instead exhibit significant difference in the photoconduction. Also, this photoconduction contrast cannot be simply attributed to the optical absorption difference between morphotropic phases[29], as detailed in the Supplementary Note 5. Therefore, there must be an external factor mediating the local photoconduction, which is not related to the intrinsic electronic properties of the morphotropic phases themselves. The only difference between the $T$-matrix phase and tilted $T'$-phase is that the $T'$-phase is epitaxially connected with the $R$-phase by the morphotropic phase boundaries. Thus, these boundaries are likely to play this role.

**Strain gradient characterization**. To explore the potential role of the phase boundaries and understand the underlying mechanism, we characterized the morphotropic phase region by high-resolution scanning transmission electron microscopy (HR-STEM) to reveal local structural features with an atomic resolution. A HR-STEM image of a $R/T'$- mixed region is shown in Fig. 3a, which demonstrates a smooth transition without defects between these morphotropic phases. This HR-STEM image is further analyzed by the geometric phase analysis (GPA) to determine local lattice deformation, i.e. strain states (see Fig. 3b, c). In comparison to the $R$-phase, the $T'$-phase possesses a larger out-of-plane lattice constant but a smaller in-plane lattice parameter, resulting in an out-of-plane tensile strain field

reaching $\varepsilon_{yy}$ ~12% and a compressive in-plane strain field as large as $\varepsilon_{yy}$ ~ −4.5% (see Fig. 3d). Note that the line profile analysis shown in Fig. 3d is performed along the area perpendicular to the phase boundaries, as marked by the red arrow in Fig. 3b. The corresponding strain gradient, i.e. the derivative of the strain with respect to the distance $l$, is shown in Fig. 3e. Clearly, there exists a giant strain gradient at the phase boundaries. Specifically, the strain gradient $\partial\varepsilon_{yy}/\partial l$ stemming from the out-of-plane lattice mismatch reaches a maximum of ~$6\times10^7$ m$^{-1}$ at the phase boundary where $R$-phase transforms into $T'$-phase whereas it peaks at ~$8\times10^7$ m$^{-1}$ at the boundary where $T'$-phase changes to $R$-phase. The larger strain gradient magnitude at the $T'$-to-$R$ phase boundary is due to its narrower transition width. Meanwhile, the strain gradient $\partial\varepsilon_{xx}/\partial l$ originating from the in-plane lattice variation reaches its negative maximum at the $R$-to-$T'$ phase boundary and a positive maximum at the $T'$-to-$R$ boundary with a magnitude of ~$4\times10^7$ m$^{-1}$. The distribution of the strain gradient $\partial\varepsilon_{yy}/\partial l$ and $\partial\varepsilon_{xx}/\partial l$ with respect to the morphotropic phases is schematically illustrated in Fig. 3f. Clearly, the strain gradients possess opposite signs at the $T'$-to-$R$ and $R$-to-$T'$ phase boundaries. In another word, the strain gradient $\partial\varepsilon_{yy}/\partial l$ is always pointing to the $T'$-phase while the $\partial\varepsilon_{xx}/\partial l$ is always pointing to the $R$-phase. This is in sharp contrast to the in-plane components of the ferroelectric polarization in these phases, which are all aligned towards the same side with respect to the phase boundaries (see Supplementary Note 6)[30]. In addition to their giant magnitude, the strain gradient preserves over a quite long distance (~4 nm) due to the smooth transition between morphotropic phases.

## Discussion

As demonstrated recently, the strain gradient has the ability to separate the photon-excited electron-hole pairs into opposite directions due to the strain gradient induced centrosymmetry breaking and associated asymmetric distribution of non-equilibrium carriers in the **k**-space[15,31]. Given the giant

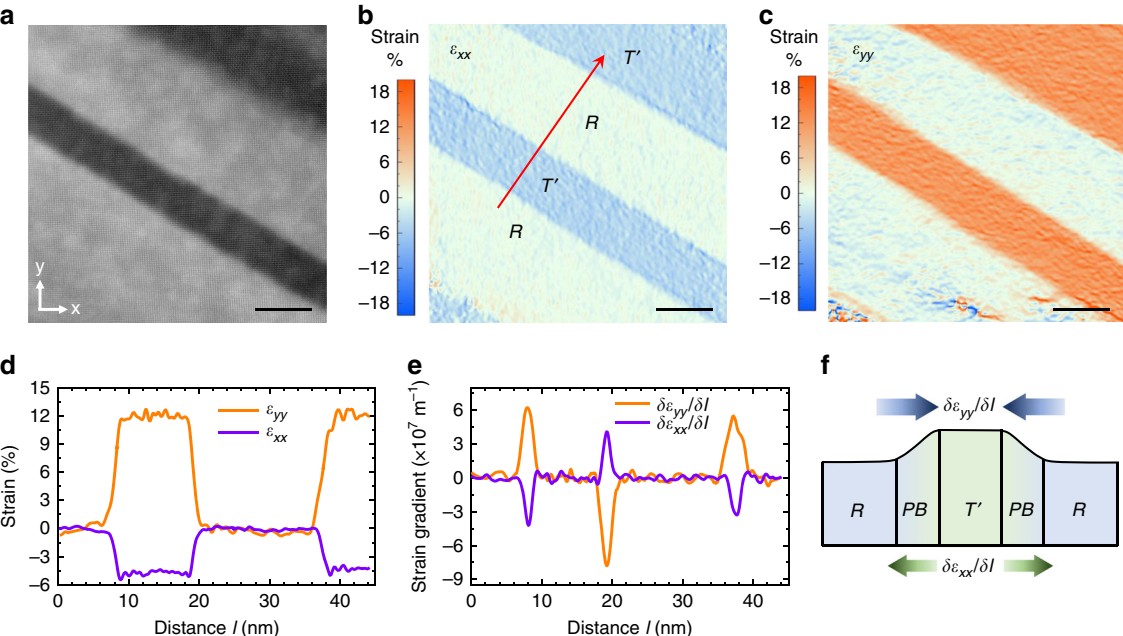

**Fig. 3** Strain and strain gradient characterization. **a** High-resolution STEM cross section image of strained BiFeO$_3$/LaAlO$_3$ thin film for the GPA analysis. Scale bar 10 nm. **b** In-plane strain ($\varepsilon_{xx}$) and **c** out-of-plane strain ($\varepsilon_{yy}$) field. Here, $x$ and $y$ directions are parallel to the LaAlO$_3$ [100]$_{pc}$ and [001]$_{pc}$ directions, respectively. **d** Strain and **e** strain gradient distribution along the direction perpendicular to the phase boundaries marked by red arrow in (**b**). **f** Schematic showing the strain gradients at the morphotropic phase boundaries. PB: phase boundary, STEM: scanning transmission electron microscopy

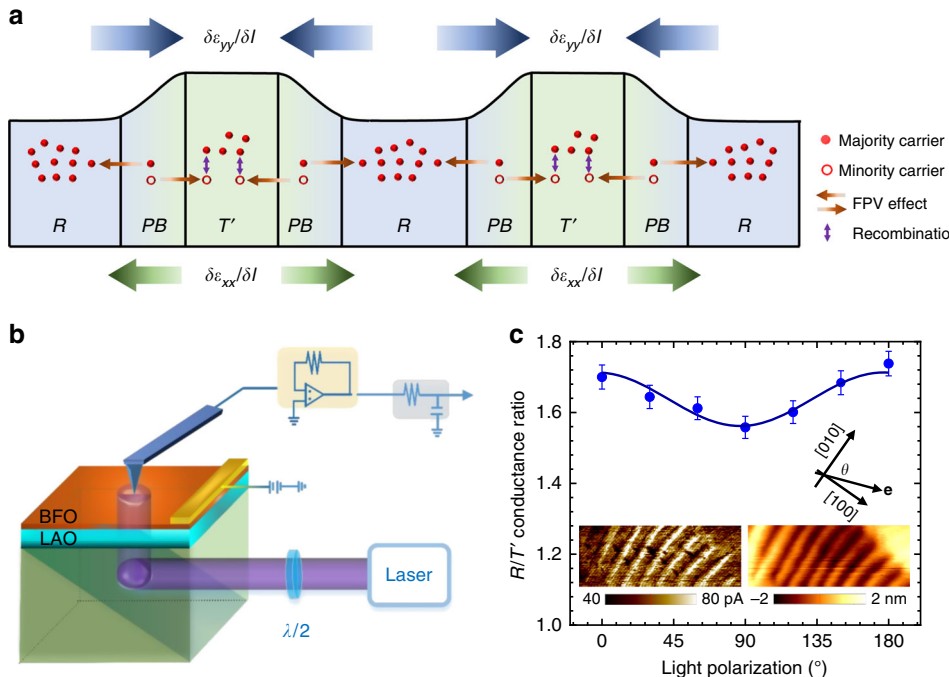

**Fig. 4** Flexo-photovoltaic effect and light polarization resolved photocurrent mapping. **a** Schematic showing the electronic process happening at the morphotropic phase boundary. The abbreviation FPV refers the flexo-photovoltaic effect. The number of the red dots denotes the resultant density of nonequilibrium carriers that determines the local photoconduction. **b** Schematic showing the illumination geometry. **c** Light polarization dependent photoconduction ratio between $R$- and $T''$-phases. The solid curve is the fit of the experimental data with Eq. 1. Inset of (**b**) shows an example of photocurrent mapping with light polarization angle of 150° (left) and its corresponding topography (right). The error bars indicate the standard deviations of photo-conductance ratio between morphotropic phases mapped in a $2 \times 0.4\ \mu m^2$ area

magnitude of the strain gradient, this unique effect (termed flexo-photovoltaic effect) is able to redistribute the non-equilibrium carriers excited at the boundaries into nearby morphotropic phases and then dramatically alter local non-equilibrium carrier density. A simple scenario can be built here. As schematically shown in Fig. 4a, the strain gradients will separate and distribute the non-equilibrium carriers, i.e. the minority carriers into the $T''$-phase and the majority carriers into the $R$-phases. Note that, despite the opposite signs of strain gradient $\partial\varepsilon_{yy}/\partial l$ and $\partial\varepsilon_{xx}/\partial l$ at the same phase boundary, their effects on redistributing non-equilibrium carriers are not necessarily compensating each other due to the entangled quantitative relation between strain gradient and flexo-photovoltaic current. The additional minority carriers distributed to the $T''$-phase will recombine the majority non-equilibrium carriers therein. This will lead to a reduced density of majority carriers and thus, supressed effective conductivity (photoconduction). On the other hand, the majority carriers delivered by the strain gradient into the $R$-phase via the flexo-photovoltaic effect would increase the majority carrier density, resulting in an enhanced photoconduction. Thanks to the large mean free path of the non-equilibrium carriers (~10–100 nm)[32], this strain gradient-modulated photoconduction would affect the whole in $R$- and $T''$-phase region, of which width is usually less than 100 nm, rather than being just limited to the area adjacent to the boundaries. Also, the relative large volume of the phase boundary regions guarantees the redistribution of a large number of non-equilibrium carriers into nearby morphotropic phases contributing to the photoconduction mediation. A semi-quantitative calculation of the flexo-photovoltaic current is given in the Methods section. It is noteworthy that the flexo-photovoltaic effect is an intrinsic property originating from the strain gradient, which is not induced by depolarization field, polarization divergence, flexo-polarization or built-in field[31,33,34]. Also, the depolarization field derived from the flexoelectric

polarization would play a minima role here, if any (see Supplementary Note 7). Parenthetically, despite these morphotropic phases themselves being intrinsic non-centrosymmetric, their intrinsic bulk photovoltaic effect is unable to tailor local photo-conductions due to the unidirectional nature of the ferroelectric polarization in each morphotropic phases. For example, the non-equilibrium carriers delivered into the $T''$-phase by the bulk photovoltaic effect of the $R$-phase located at the left side of the $T''$-phase would be extracted out of the $T''$-phase by the $R$-phase on the right side, resulting in a null effect (see Supplementary Note 6).

A fingerprint feature of the flexo-photovoltaic effect is its dependence on the incident light polarization, which can be expressed as (see "Methods"):

$$I_{FPV} = I_0 A[B_1 + B_2\cos(2\theta + \varphi)] \qquad (1)$$

where $I_{FPV}$ is the short-circuit current generated by the flexo-photovoltaic effect, $I_0$ is the light intensity, $A$ is the morphotropic phase boundary area, $B_1$ and $B_2$ are the effective flexo-photovoltaic coefficient, $\theta$ is the angle made between light polarization and the LaAlO$_3$ [010]$_{pc}$ direction (see the inset of Fig. 4c) and $\varphi$ is the offset angle. Accordingly, the amount of non-equilibrium carriers separated by the strain gradient at the phase boundaries would also depend on the light polarization. Consequently, the photoconduction of $T''$- and $R$-phase, which is mediated by the strain gradient, can be tailored by the incident light polarization as well. To validate this scenario, we mapped the photocurrent contrast between $T''$- and $R$-phases in a mixed phase region by using the illumination strategy depicted in Fig. 4b, which enables rotation of the light polarization by a half-wavelength plate (see "Methods"). The back side of the LaAlO$_3$ substrates has been polished to allow the light transmission and illumination in a perpendicular geometry. Due the large band gap of the LaAlO$_3$ substrates ($E_g = 5.6$ eV), absorption of the 405 nm

light would only happen in the strained $BiFeO_3$ films. As shown in Fig. 4c, the photoconduction contrast between the $R$-phase and the $T'$-phase, i.e. the ratio between maxima and minima of the photocurrent in mixed phase region, shows a distinct dependence on the light polarization. The photocurrent contrast reaches a maximum when the light polarization runs parallel to the morphotropic phase boundaries whereas it decreases to a minimum value when the light polarization is perpendicular to the boundaries. Also, the light polarization dependence of the photoconductance ratio between the morphotropic $T'$-, $R$-phase and matrix $T$-phase is given in Supplementary Note 8. More importantly, the variation of these photoconduction ratio therein can be well fitted by Eq. 1. Due to the small variation amplitude of the photoconduction ratio between $T'$ and $T$ phase, the photoconduction ratio between $R$-phase and the $T'$-phase can also be well fitted by Eq. 1 (see Fig. 4c). Despite that the morphotropic phases might have a certain light polarization dependent optical absorption due to the tilting of ferroelectric polarization away from surface normal direction (for more details see ref. [35]), their light polarization dependent photoconduction cannot account for the sinusoidal dependence shown in Fig. 4c (see Supplementary Note 5). Therefore, this clearly demonstrates that it is the flexo-photovoltaic effect arising from the strain gradient at the phase boundaries that mediates the local photoelectric properties. Also, this indicates the flexo-photovoltaic effect in the morphotropic phase boundary maximizes when light polarization is parallel to the boundary whereas it minimizes when light polarization is perpendicular to the boundary.

We can now explain the scattered effects of the morphotropic phase boundaries on the photo sensitivity of strained $BiFeO_3$ films with different electrode geometries. Due to the mediation of the flexo-photovoltaic effect, the $R$-phase possesses an enhanced conduction while the tilted $T'$-phase shows a suppressed conduction under illumination. In the out-of-plane capacitor geometry, the top and bottom electrodes are directly shunted by the highly conductive $R$-phases, facilitating the current conduction under illumination. On the contrary, the gap between the electrodes in the coplanar geometry is in the order of micrometres, which is much larger than the size of the morphotropic phases. The current conduction path between the electrodes consists of the $T$- matrix, $T'$-, and $R$-phases connecting in a series manner. The final conduction of the coplanar devices is thus determined by the parts of highest resistance, i.e. the tilted $T'$-phase. Therefore, the photoconduction of the coplanar device would decrease with increasing the density of the morphotropic $T'$-phase and the density of the morphotropic phase boundaries. This also indicates that the strain gradient not only modulates local electronic properties but also becomes an effective stimulus to control the overall performance of the photoelectric devices on a macroscopic level.

In summary, we locally characterized the electronic properties of morphotropic phases in the strained $BiFeO_3$ thin films in both dark and illuminated conditions with a nanometre resolution. In dark condition, the morphotropic $R$-phase exhibits a supressed conductivity compared to the tilted $T'$-phase. Under illumination, the photoconduction of the $R$-phase is significantly enhanced. However, the tilted $T'$-phase shows negative photoconductivity-like feature compared to the $T$-matrix with the same chemical component and lattice structure. We explain this unexpected behaviour by the contribution of flexo-photovoltaic effect at the morphotropic phase boundaries, where the strain gradient is very large.

## Methods

**Thin film growth and measurement devices.** The strained $BiFeO_3$ thin films were grown on $LaAlO_3(001)$ substrates by pulsed laser deposition. To map the dark

current distribution through the conventional cAFM system, a $La_{0.7}Sr_{0.3}MnO_3$ layer with 5 nm thickness was deposited as bottom electrode for the strained $BiFeO_3$ film. To characterize the photocurrent distribution, an in-plane platinum electrode was evaporated on the $BiFeO_3/LaAlO_3$ thin film to complete the current circuit. A side electrode is preferred than the bottom electrode for the photocurrent mapping by the Ph-AFM system as it allows an Ohmic-like $I$-$V$ characteristic of the tip/$BiFeO_3$ system. The $BiFeO_3$ films were grown at 640 °C with an oxygen pressure of 0.15 mbar. The laser energy fluence was set as 1 j cm$^{-2}$ and frequency was 5 Hz. The $La_{0.7}Sr_{0.3}MnO_3$ layer was grown at 600 °C with an oxygen pressure of 0.15 mbar, laser fluence of 1 j cm$^{-2}$ and a frequency of 5 Hz.

**Dark conduction and photoconduction mapping.** The spatially resolved dark conduction distribution is mapped by the XE-100 Park AFM system equipped with a home built current amplifier/filter system. During the dark current measurement, a small bias (2 V) is applied to the $La_{0.7}Sr_{0.3}MnO_3$ layer, which would not trigger the phase transition between $R$ and $T$ phases. The photocurrent distribution is mapped by the Ph-AFM system consisting of an AFM-based system (XE-100, Park) modified by a custom current amplifier (Femto, DLPCA-200) /filter system (Stanford Research Systems, SR560) and an optical system. The NSC14/Pt AFM tip (Mikromasch) used in this work has a diameter less than 25 nm, which enables high lateral resolution in both topography and current mapping. The optical system allows illumination on the $BiFeO_3$ surface with a polarized $\lambda = 405$nm light (3.06eV). The linear light polarization can be continuously rotated via a half-wavelength plate. During the photocurrent mapping, the light intensity was set as 1 W cm$^{-2}$ and a bias of 2 V was applied to the side Pt electrode.

**Light polarization resolved photocurrent mapping.** To characterize the light polarization dependence of local photoconduction, we mapped the spatially resolved photocurrent distribution of the same area containing morphotropic $R$- and $T'$-phase and matrix $T$-phase under illumination with various light polarization direction. To mitigate measurement errors and uncertainties, the photoconduction of each phases at a certain light polarization is obtained by averaging the photocurrent acquired in scanned area. Note that, apart from the light polarization direction, the magnitude of the photocurrent acquired during scanning processes is also vulnerable to other external factors, such as the electrical contact quality between the conductive tip and sample surface. To make our measurements more robust, we exploited a photoelectric feature of the matrix $T$-phase, i.e., the independence of its photoconduction on the light polarization (see Supplementary Fig. 7). Accordingly, we used the photocurrent obtained on the matrix $T$-phase as the baseline for each scan and normalize the photocurrent of $R$- and $T'$-phase to that of matrix $T$-phase. This mitigates the extrinsic variations and reveals an intrinsic correlation of local photoelectric properties on the light polarization direction. We prefer this dynamic measurement method to the fixed-point method as the latter would induce substantial error due to the thermal drift, small dimension of those morphotropic phase, etc.

**Scanning transmission electron microscopy.** TEM specimens were made using standard focussed ion beam (FIB) lift-out procedures on a JEOL JIB-4500 FIB-SEM. All STEM images were taken using a double CEOS corrected (to third order), Schottky emission JEOL ARM-200F microscope operating at 200 kV in STEM mode. Sample drift and scan distortions were mitigated by aligning orthogonal scan pairs[36]. GPA measurements were performed with an in-house developed software Strain++ (http://jjpeters.github.io/Strainpp/) using the 001 and 010 reference vectors and Gaussian masks with $\sigma = 0.87$ nm$^{-1}$[37]

**Calculation on the redistribution of non-equilibrium carriers by the flexo-photovoltaic effect.** The TEM characterization reveals a giant strain gradient with a magnitude reaching $8 \times 10^7$ m$^{-1}$ at the morphotropic the phase boundaries. The strain gradient induced electrical polarization (flexoelectric effect), i.e. non-centrosymmetry, can be numerically estimated. Although the flexoelectric coefficient of the strain $BiFeO_3$ films has not yet been measured, it would be reasonable to assume its magnitude similar order of magnitude as other perovskite oxide, such as $SrTiO_3$ and $BaTiO_3$ ($f\sim3 \times 10^{-9}$ C m$^{-1}$)[38]. Thus, according to the flexoelectric effect:

$$P = f\frac{\partial \varepsilon}{\partial x} \qquad (2)$$

The strain gradient induced electrical polarization is about $P = 24$ μC cm$^{-2}$, which is similar to the in-plane ferroelectric polarization of $BiFeO_3/LaAlO_3$ films[39]. This indicates the giant strain gradient is able to generate substantial non-centrosymmetry comparable to that of perovskite ferroelectric materials.

To semi-quantitatively calculate the ability of the flexo-photovoltaic effect manifested at the phase boundaries in the mediation of local photoconduction, a simple method is to compare the density of nonequilibrium carriers generated in the $T$-phase matrix and the density of carriers redistributed by the flexo-photovoltaic effect. The density of nonequilibrium carriers in the $T$-phase matrix can be estimated from its photoconductivity measured by in-plane electrodes ($\sigma_{ph} = 1 \times 10^{-5}$ S cm$^{-1}$)[17]. Given the carrier mobility as 1.5 cm$^2$ V$^{-1}$ s$^{-1}$, the light generated nonequilibrium carrier density is $n_{ph} = 4.2 \times 10^{13}$ cm$^{-3}$. Similar to the

bulk photovoltaic effect, the current density $J_{FPV}$ generated by the flexo-photovoltaic effect can be predicted with following equation:

$$J_{FPV} = G\alpha I_0 \qquad (3)$$

where $G$ is the Glass coefficient, $\alpha$ light absorption coefficient ($\sim 1 \times 10^5 \, cm^{-1}$) and $I_0$ is the light intensity ($1 \, W \, cm^{-2}$)[40]. Given the giant non-centrosymmetry induced by the strain gradient, it would be reasonable to assume a quasi-Glass coefficient of the flexo-photovoltaic effect similar to other ferroelectric perovskite oxides ($\sim 10^{-9} \, cm \, V^{-1}$). Thus, the flexo-photovoltaic current density is $0.1 \, mA \, cm^{-2}$. Under the steady-state condition, the carriers redistributed by the flexo-photovoltaic effect would built up a space charge field, preventing further carrier redistribution. This field can be calculated by:

$$E_{FPV} = \frac{J_{FPV}}{\sigma_{ph}} \qquad (4)$$

By using the values calculated, the field $E_{FPV} = 10 \, V \, cm^{-1}$. The carrier accumulated at each side of the morphotropic phase boundaries is give as

$$Q = CE_{FPV}d = \varepsilon_0\varepsilon_r A E_{FPV} \qquad (5)$$

where $C$ is the capacitance, $d$ is the thickness of the phase boundary, $A$ is the area of the phase boundary, $\varepsilon_r$ is the dielectric constant ($\sim 50$). Therefore, the averaged carrier density $n_{FPV}$ redistributed by the flexo-photovoltaic effect into morphotropic phases can be given as:

$$n_{FPV} = \frac{Q}{V} = \frac{\varepsilon_0\varepsilon_r E_{FPV}}{l} \qquad (6)$$

where $l$ is the lateral length of the morphotropic phases ($l\sim 100 \, nm$). Thus, the carrier density redistributed by the flexo-photovoltaic effect is $n_{FPV}\sim 3 \times 10^{13} \, cm^{-3}$, which is similar to that of non-equilibrium carrier density $n_{ph}$ in the $T$ phase matrix. Therefore, the flexo-photovoltaic effect is able to dramatically modulate local carrier density and, thus, photoconduction. This analysis is also confirmed by our recent work. The strain gradient induced by the AFM tip, which is in the similar range of that manifests at the phase boundary, generates substantial flexo-photovoltaic current[15]. Therefore, the strain gradient manifested at the morphotropic phase boundaries is sufficient to produce the claimed flexo-photovoltaic effect to mediate local photoconduction.

**Calculation of the light polarization dependent flexo-photovoltaic effect.** The flexo-photovoltaic effect in the morphotropic phase boundaries can be effectively expressed by a third-rank tensor $\beta_{ijk}^{eff}$[31]:

$$\beta_{ijk}^{eff} = \begin{pmatrix} \beta_{11} & \beta_{12} & \beta_{13} & \beta_{14} & \beta_{15} & \beta_{16} \\ \beta_{21} & \beta_{22} & \beta_{23} & \beta_{24} & \beta_{25} & \beta_{26} \\ \beta_{31} & \beta_{32} & \beta_{33} & \beta_{34} & \beta_{35} & \beta_{36} \end{pmatrix} \qquad (7)$$

The incident light along the surface normal direction (i.e. $z$ direction) can be expressed as:

$$\mathbf{e} = \begin{pmatrix} \cos\theta & \sin\theta & 0 \end{pmatrix} \qquad (8)$$

where $\alpha$ is the angle between the light polarization and LaAlO$_3$ [010]$_{pc}$ direction. Note that the coordinate $x$ is set parallel to the LaAlO$_3$ [010]$_{pc}$ direction and $y$ is parallel to the [010]$_{pc}$ direction. The $\theta$ is the angle made between the light polarization and LaAlO$_3$ [010]$_{pc}$ direction (i.e. $x$ axis). Then, the light polarization field $e_je_k$ equals to

$$e_je_k = \begin{pmatrix} \cos^2\theta & \sin^2\theta & 0 & 0 & 0 & 2\sin\theta\cos\theta \end{pmatrix}^T \qquad (9)$$

In the case of the morphotropic phase boundary, the flexo-photovoltaic current perpendicular to the interface, namely, the current along the [100]$_{pc}$ direction in Fig. 4c, plays the major role in mediating local photoconduction. Thus, the flexo-photovoltaic current $I_{FPV}$ can be given as:

$$I_{FPV} = I_0 A \beta_{2jk}^{eff} e_j e_k$$

$$= I_0 \begin{pmatrix} \beta_{21} & \beta_{22} & \beta_{23} & \beta_{24} & \beta_{25} & \beta_{26} \end{pmatrix} \begin{pmatrix} \cos^2\theta \\ \sin^2\theta \\ 0 \\ 0 \\ 0 \\ 2\sin\theta\cos\theta \end{pmatrix} \qquad (10)$$

$$= I_0 A \left( \beta_{21}\cos^2\theta + \beta_{22}\sin^2\theta + 2\beta_{26}\sin\theta\cos\theta \right)$$

$$= I_0 A [B_1 + B_2\cos(2\theta + \varphi)]$$

where $B_1 = \frac{1}{2}\beta_{21} + \frac{1}{2}\beta_{22}$, $B_2 = \sqrt{\frac{1}{4}\left(\beta_{21} - \beta_{22}\right)^2 + \beta_{26}^2}$, $\varphi$ is the offset angle, $I_0$ is the light intensity, $A$ is the morphotropic phase boundary area. The offset angle $\varphi$ is fitted to be 6° in Fig. 4c.

## Data availability

The data that support the findings of this study are available at the University of Warwick open access research repository (http://wrap.warwick.ac.uk/116941).

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

## Acknowledgements

M.A. acknowledges the Wolfson Research Merit and Theo Murphy Blue-sky Awards of Royal Society. The work was partly supported by the EPSRC (UK) through grants no. EP/M022706/1, EP/P031544/1 and EP/P025803/1. The authors acknowledge Hangbo Zhang for the schematic drawing.

## Author contributions

M.M.Y. and M.A. conceived the idea and designed the experiments. M.M.Y. and A.N.I. prepared the thin films. M.M.Y. performed the local electrical characterization. J.J.P.P. and A.M.S. performed the TEM measurement. M.M.Y. wrote the manuscript and M.A. A.N.I., J.J.P.P. revised the manuscript. All the authors contributed to the discussion.

## Additional information

**Competing interests:** The authors declare no competing interests.

**Peer Review Information:** *Nature Communications* thanks the anonymous reviewers for their contribution to the peer review of this work. Peer reviewer reports are available.

