## [Peer Review File · Nature Communications]

Reviewers' comments:

Reviewer #1 (Remarks to the Author):

In this report, the authors demonstrate that large strain gradients can be considered responsible for the local photoconduction in 'mixed phase' regions in popular multiferroic BiFeO₃ thin films grown on LaAlO₃ (001) substrate. It is shown using conductive tip atomic force microscopy scans that in dark conditions, maxima in the conductivity appear at the tilted T' regions, whereas when the sample is illuminated by light, the maxima in conductivity are shifted towards the relaxed R' regions. Taking into account the large strain gradient that exists between these mixed phase regions, the authors explain their results using the recently shown flexo-photovoltaic effect. The work is well presented, the data quality is quite good (except for one comment as shown below), and the story is well explained. This paper in my opinion is suitable for the readership of Nature Communications, it opens up new perspectives for light-controlled electronic devices; however before publication some minor errors and a possible interpretation of the data should be ruled out, as explained in the following.

In Figure 1, the scale bar appears to read 'μm' – I would assume that it is meant to read 'nm'. Also, regarding the topographic scan in (a) – the image quality is rather poor – perhaps this is due to the the c-AFM tip? Can the authors improve the quality of this scan? Also, in part (b) – the red arrow to indicate the location of the profile is not very clear. Perhaps a different color would be better.

A concern in the paper is that the authors do not take into account that the T' and R' phases have quite different band gaps (about 300 meV between them). If we consider the published absorption coefficient by Chen et al (APL 2010, 96, 131907), then at 3.06 eV (the light wavelength used here) the absorption coefficient of the T' phase is $\alpha_{T'} \sim 1e5 /cm$; while for the R' phase it is $\alpha_{R'} \sim 2.5e5 /cm$. This is a more than two times difference, which would imply that under illumination the various regions of the films would be absorbing light at different rates, which could possibly mean that the steady-state conditions (which is presumably how the films are measured by c-AFM) would involve different populations of carriers from the various phases. Could it be that the space charge field thus induced through this spatially-inhomogeneous generation of photocharges could be the explanation for the observed phenomena?

Although unlikely, the local variations in the photo-generated carriers, and thus different space charge fields generated, could be the reason for the observed results. Regarding Figure 4, where there is a dependence on the light polarization, it is important to recall that the two BFO phases show quite a strong dependence of their absorption on the light polarization direction [see Schmidt et al., PRB 92, 075310 (2015)]. Also, we must remember that the symmetry of the T' phase is almost certainly MC, while the symmetry of the R' phase is probably monoclinic MA. Importantly these two phases will have their optical axes rotated by 45 degree relative to each other, in the plane – this would mean that changing the light polarization would probe various (and differing) proportions of the various complex refractive indices (and thus optical absorption) of the two phases. As mentioned above, although this scenario is unlikely; it would be appreciated that the authors to conduct some kind of analysis (using the papers given above as reference works) and justification that such a situation can be ruled out.

Reviewer #2 (Remarks to the Author):

This is an interesting paper on a currently important topic that deserves to be considered for publication in NC.

At this point, I have one general issue with the paper that I hope authors will be able to address. The authors show distribution of strain gradient across the film and correlate this strain gradient to the variation in photo-current. I feel it is essential to treat the problem numerically and show that the estimated/measured strain gradient is indeed sufficient to produce the claimed flexo-

photovoltaic effect.

It would also be very helpful to show a schematic of strain gradient, photo current and topography across an area of the film (a combination of Fig. 1c, 2c and 3e). This can be a conceptual drawing included in SI.

Please include information on GPA analysis either in SI or in Methods.

Reviewer #3 (Remarks to the Author):

Using atomic probe microscopy techniques, authors performed an investigation on the photoelectric properties in R/T' mixed phase and pure T matrix regions. Authors claimed that the photoconductance in R phase region is enhanced and is reduced in T' phase and it is reversed behavior considering the dark current. However, I think there are some critical issues to be resolved to justify the claims as listed below.

Even if the claims are correct, I doubt that it is worth for publication in Nature Communications which requires a significant breakthrough in the field. In my opinion, the study is just an extension of some of the previous works considering that the separation of light-induced electron-hole pair at the morphotropic phase boundary (MPB), mediated by flexoelectricity due to the huge strain gradient, is already well demonstrated from the work in Nat. Nanotech. 10, 972 (2015). Also, the direction of the strain gradient at both MPB in the mixed phase region is already reported in the work in Adv. Func. Mater. 25, 3405 (2015). Although the study contains some interesting observations, I do not recommend the paper for the publication in Nature Communications.

(The main issues)

(1) It is important to mention what kind of tip was used (correspondingly the size of the tip radius) since the lateral dimension of the R/T' mixed region is ~100nm and it could be comparable or even smaller to the size of the radius for some AFM tips. It seems that height difference between R/T' boundary and T'/R boundary is less than 1nm which is much smaller than the reported value for BFO/LAO with a similar film thickness in previous papers (e.g. Science 326, 977 (2009), JAP 112, 064102 (2012)). This is usually the signature for using a tip with a large radius. If it is the case, the contact region between the tip and the sample could be not just below the centre of the tip, and thus, it is difficult to determine the exact location of the current maxima or minima from the line profile in the manuscript.

(2) Authors claimed that R-phase matrix has a lower conductivity and tilted T' phase has a higher conductivity compared to one with T-phase region in the dark condition. If conductivity is only dependent on the type of structure and not influenced by conduction property in morphotropic phase boundary, why the currents on regions with tilted T' phase or with R phase are not uniform even in the regions away from the morphotropic phase boundary?

(3) Authors excluded the contribution of a bulk-photovoltaic effect considering its unidirectional nature by presenting schematic illustration in Fig. S3. However, I don't believe that the in-plane polarization in the mixed phase region should be unidirectional as schematically drawn in Fig. S3. The authors didn't present any in-plane PFM data and just assume the ideal unidirectional polarization distribution, meanwhile other works on the mixed-phase BFO show very complicated polarization distribution.

(4) Since all the experimental data shown in this work are under the application of electric bias, the effect could be due to the combination of the photoconductivity and photovoltaic effects, yielding a difficulty for complete understanding of the observed phenomena.

(5) What are the majority carrier and minority carrier? The direction of the depolarization field due

to the flexo-polarization should be toward R-phase, thus, electrons would move to the T' phase region and holes to the R phase region. In this sense, the majority carrier should be the hole. Is this correct?

(minor issues)

(1) The scale bar in Fig 1a is wrong.

(2) The uppermost line for Fig 2c is not along with the ones for Fig 2a,b unlike the case in Fig 1.

(3) "Fig. 3b" should be changed to "Fig. 2b" in line 116 of the manuscript for publication in its present form.

Response to Reviewers' comments

“Strain-Gradient Mediated Local Conduction in Strained BiFeO₃ Film” by Ming-Min Yang *et al.*, Manuscript No. NCOMMS-18-34631.

We would like to thank the reviewers for the thorough assessment of our work. Below, we have listed the issues/comments raised by the reviewers and our corresponding replies. We believe that we addressed all the issues raised by the Reviewers, thus making the message delivered by our manuscript (MS) clearer for readers.

Response to Reviewer #1

Question 1.1 *In Figure 1, the scale bar appears to read ‘ μm ’ – I would assume that it is meant to read ‘nm’. Also, regarding the topographic scan in (a) – the image quality is rather poor – perhaps this is due to the the c-AFM tip? Can the authors improve the quality of this scan? Also, in part (b) – the red arrow to indicate the location of the profile is not very clear. Perhaps a different color would be better.*

Answer 1.1 The scale bar in Figure 1a has been changed to ‘nm’ instead. The quality of (a) is affected by a convolution of complicated features of the morphotropic phases in strained BiFeO₃ thin film. As suggested by the Reviewer, we have improve the image quality to some extent by enhancing the contrast. We have also moved the arrow from part (b) to part (a).

Action Taken 1.1 The scale bar unit has been corrected to ‘nm’ and the image quality of (a) has been improved. The arrow has been moved to part (a).

Question 1.2 *A concern in the paper is that the authors do not take into account that the T' and R' phases have quite different band gaps (about 300 meV between them). If we consider the published absorption coefficient by Chen et al (APL 2010, 96, 131907), then at 3.06 eV (the light wavelength used here) the absorption coefficient of the T' phase is $\alpha_{T'} \sim 1e5$ /cm; while for the R' phase it is $\alpha_{R'} \sim 2.5e5$ /cm. This is a more than two times difference, which would imply that under illumination the various regions of the films would be absorbing light at different rates, which could possibly mean that the steady-state conditions (which is presumably how the films are measured by c-AFM) would involve different populations of carriers from the various phases. Could it be that the space charge field thus induced through this spatially-inhomogeneous*

generation of photo-charges could be the explanation for the observed phenomena? Although unlikely, the local variations in the photo-generated carriers, and thus different space charge fields generated, could be the reason for the observed results.

Answer 1.2 We thank the Reviewer for bringing the light absorption issue into discussion. Indeed, there is difference of a factor of two in the absorption coefficients of the strain-free rhombohedral BiFeO₃ grown on DyScO₃ substrate and the tetragonal phase grown in LaAlO₃ phase, according to Chen's work [APL 2010, 96, 131907]. However, it is necessary to clarify that the morphotropic *R* phase in the mixed phase region of the strained BiFeO₃/LaAlO₃ thin film is distinct from the typical rhombohedral BiFeO₃ phases. Although, for the sake of simplicity, we termed it in the manuscript as *R*-phase, the morphotropic *R*-phase is not the strain-free rhombohedral BiFeO₃ [Chen *et al. Adv. Funct. Mater.* **21**, 133 (2011); Damodaran *et al. Adv. Mater.* **23**, 3170 (2011); Chen *et al. Phys. Rev. B* **84**, 094116(2011)]. Thus, it is not suitable to compare the bandgap and light absorption coefficient between the morphotropic phases in the strained BiFeO₃ films by using the values from the rhombohedral BiFeO₃/DyScO₃ thin film. An alternative way to get an insight into the bandgap values and light absorption coefficients of the morphotropic phases in the BiFeO₃/LaAlO₃ film is to study the thickness dependent absorption coefficient, as the volume of the *R* phase increases with the thickness. In this regard, our previous optical studies reveals a similar bandgap values and light absorption coefficients on a thin BiFeO₃/LaAlO₃ film, which consists of pure tetragonal-like *T* phase, and a thick BiFeO₃/LaAlO₃ film wherein the mixed morphotropic phases accounts for 50% area [Himcinschi *et al. APL* **106**, 012908(2015)]. This indicates a negligible difference of bandgap ($\Delta E_g \leq 50$ meV) and absorption coefficient between the morphotropic phases in the strained BiFeO₃ films. As a results, the non-equilibrium carriers density induced by light should be similar in both *R*-phase and *T*-phase morphotropic phases of the strained BiFeO₃ films. Thus, the large contrast of the photoconduction between the phases cannot stem in the light absorption difference.

Action Taken 1.2 The sentence below has been added to the page 7 of the manuscript marked as red: "Also, given the negligible differences of bandgap and light absorption coefficient between the *R*- and *T*-phase, the incident light would generate similar density of non-equilibrium carriers in the morphotropic phases."

Question 1.3 Regarding Figure 4, where there is a dependence on the light polarization, it is important to recall that the two BFO phases show quite a strong dependence of their absorption on the light polarization direction [see Schmidt *et al., PRB* **92**, 075310 (2015)]. Also, we must

remember that the symmetry of the T' phase is almost certainly M_C , while the symmetry of the R' phase is probably monoclinic M_A . Importantly these two phases will have their optical axes rotated by 45 degree relative to each other, in the plane – this would mean that changing the light polarization would probe various (and differing) proportions of the various complex refractive indices (and thus optical absorption) of the two phases. As mentioned above, although this scenario is unlikely; it would be appreciated that the authors to conduct some kind of analysis (using the papers given above as reference works) and justification that such a situation can be ruled out.

Answer 1.3 The non-centrosymmetric materials generally possess anisotropic light absorption depending on the light polarization direction with respect to the optical axis. As reported by Schmidt *et al.*, the strained BiFeO₃/LaAlO₃ (001) thin film shows a uniaxial anisotropic absorption coefficients ($\alpha_a = \alpha_b \neq \alpha_c$) [*Phys. Rev. B* **92**, 075310(2015)]. Accordingly, the light absorption coefficient of a linear polarized light in the strained BiFeO₃ film would remain a constant value independent of the light polarization direction if the light incidents along the out-of-plane direction (i.e. *c* direction). This is the strategy we used in our work to keep the light absorption unchanged while rotating the light polarization direction (see Figure 4a). Note that, as mentioned in **Answer 1.2**, the optical properties of the morphotropic *R* phase would be similar to that of the *T'* phase, rather than to the rhombohedral BiFeO₃ film grown in SrTiO₃ or DyScO₃ substrates. Also, the monoclinic distortion of the *R* phase ($0 \pm 0.3^\circ$) is even smaller than *T* phase ($\sim 1.9^\circ$), indicating that the *R* phase is likely to possess the uniaxial optical properties ($\alpha_a = \alpha_b \neq \alpha_c$). Therefore, by illuminating the strained BiFeO₃ thin film along the out-of-plane direction with a help of the prime mirror, the light absorption in both morphotropic *R* and *T* phases remain a constant value.

Action Taken 1.3 The following sentence has been added to the page 11 of the manuscript marked by red: “Due to the uniaxial optical properties of the strained BiFeO₃ thin films, i.e. the light absorption coefficient $\alpha_a = \alpha_b \neq \alpha_c$, the light absorption in the film would remain constant while rotating the light polarization angle.”

Response to Reviewer #2

Question 2.1 *This is an interesting paper on a currently important topic that deserves to be considered for publication in NC. At this point, I have one general issue with the paper that I hope authors will be able to address. The authors show distribution of strain gradient across the film*

and correlate this strain gradient to the variation in photo-current. I feel it is essential to treat the problem numerically and show that the estimated/measured strain gradient is indeed sufficient to produce the claimed flexo-photovoltaic effect.

Answer 2.1 We appreciate the Reviewer's positive comment on our work. It is a good suggestion to analyze numerically whether the strain gradient manifested at the morphotropic phase boundaries is large enough to redistribute the light-generated non-equilibrium carriers.

The TEM characterization reveals a giant strain gradient with a magnitude reaching $8 \times 10^7 \text{ m}^{-1}$ at the morphotropic the phase boundaries. The strain gradient induced electrical polarization (flexoelectric effect), i.e. non-centrosymmetry, can be numerically estimated. Although the flexoelectric coefficient of the strained BiFeO₃ films has not yet been measured, it would be reasonable to assume its magnitude similar order of magnitude as the other perovskite oxide, such as SrTiO₃ and BaTiO₃ ($f \sim 3 \times 10^{-9} \text{ C/m}$) [Zubko *et al.*, *Phys. Rev. Lett.* **99**, 167601(2007); Narvaez *et al.*, *Nature* **538**, 219(2016)]. Thus, according to the flexoelectric effect:

$$P = f \frac{\partial \varepsilon}{\partial x}$$

The strain gradient induced electrical polarization is about $24 \mu\text{C}/\text{cm}^2$, which is similar to the in-plane ferroelectric polarization of BiFeO₃/LaAlO₃ films [Chen *et al.*, *Phys. Rev. B* **86**, 235125(2012)]. This indicates the giant strain gradient is able to generate substantial non-centrosymmetry comparable to that of perovskite ferroelectric materials.

To semi-quantitatively estimate the ability of the flexo-photovoltaic effect manifested at the phase boundaries in the mediation of local photoconduction, a simple method is to compare the density of non-equilibrium carriers generated in the *T*-phase matrix and the density of carriers redistributed by the flexo-photovoltaic effect. The steady-state density of non-equilibrium carriers in the *T*-phase matrix can be estimated from its photoconductivity measured by in-plane electrodes ($\sigma_{ph} = 1 \times 10^{-5} \text{ S} \cdot \text{cm}^{-1}$) [Bhatnagar *et al.*, *Nano Lett.* **14**, 5224(2014)]. Given the carrier mobility as $1.5 \text{ cm}^2 \cdot \text{V}^{-1} \cdot \text{s}^{-1}$, the light generated non-equilibrium carrier density is $n_{ph} = 4.2 \times 10^{13} \text{ cm}^{-3}$. Similar to the bulk photovoltaic effect, the current density J_{FPV} generated by the flexo-photovoltaic effect can be predicted with following equation:

$$J_{FPV} = G\alpha I_0$$

where G is the Glass coefficient, α light absorption coefficient ($\sim 1 \times 10^5 \text{ cm}^{-1}$) and I_0 is the light intensity ($1 \text{ W}/\text{cm}^2$) [Glass *et al.* *Appl. Phys. Lett.* **25**, 233(1974)]. Given the giant non-centrosymmetry induced by the strain gradient, it would be reasonable to assume a quasi-Glass

coefficient of the flexo-photovoltaic effect similar to that of other ferroelectric perovskite oxides ($\sim 10^{-9} \text{ cm} \cdot \text{V}^{-1}$). Thus, the flexo-photovoltaic current density is $0.1 \text{ mA} \cdot \text{cm}^{-2}$. Under the steady-state condition, the carriers redistributed by the flexo-photovoltaic effect would built up a space charge field, preventing further carrier redistribution. This field can be calculated by:

$$E_{FPV} = \frac{J_{FPV}}{\sigma_{ph}}$$

By using the calculated values, the field $E_{FPV} = 10 \text{ V} \cdot \text{cm}^{-1}$. The carrier accumulated at each side of the morphotropic phase boundaries is give as

$$Q = CE_{FPV}d = \varepsilon_0\varepsilon_rAE_{FPV}$$

where C is the capacitance, d is the thickness of the phase boundary, A is the area of the phase boundary, ε_r is the dielectric constant (~ 50). Therefore, the averaged carrier density n_{FPV} redistributed by the flexo-photovoltaic effect into morphotropic phases can be given as:

$$n_{FPV} = \frac{Q}{V} = \frac{\varepsilon_0\varepsilon_rE_{FPV}}{l}$$

where l is the lateral length of the morphotropic phases ($l \sim 100 \text{ nm}$) and V is the volume of morphotropic phases ($V = Al$). Thus, the carrier density redistributed by the flexo-photovoltaic effect is $n_{FPV} \sim 3 \times 10^{13} \text{ cm}^{-3}$, which is similar to that of non-equilibrium carrier density n_{ph} in the T phase matrix. Therefore, the flexo-photovoltaic effect is able to dramatically modulate local carrier density and, thus, photoconduction. This analysis is also confirmed by our recent work wherein the strain gradient induced by the AFM tip, which is in the similar range of that manifests at the phase boundary, generates substantial (flexo-)photovoltaic current [Yang *et al.*, *Science* **360**, 904(2018)]. Therefore, the strain gradient manifested at the morphotropic phase boundaries is sufficient to produce the claimed flexo-photovoltaic effect.

Action taken 2.1 The analysis described above has been added into Supplementary Information following Figure S6. And following sentence has been added to the manuscript: “Given the giant magnitude of the strain gradient, this unique effect (termed the flexo-photovoltaic effect) is able to redistribute the non-equilibrium carriers excited at the boundaries into nearby morphotropic phases and then dramatically alter local non-equilibrium carrier density (see Fig. S6).”

Question 2.2 *It would also be very helpful to show a schematic of strain gradient, photocurrent and topography across an area of the film (a combination of Fig. 1c, 2c and 3e). This can be a conceptual drawing included in SI.*

Answer 2.2 It is a good idea to show a conceptual schematic in the SI to elucidate the electronic process happening at the morphotropic phases under illumination. The schematic shown in Figure R1.

Figure R1. Schematic showing the electronic process happening at the morphotropic phase boundary with the variation of local non-equilibrium carrier density. The abbreviation FPV refers the flexo-photovoltaic effect. The number of the red dots denotes the density of non-equilibrium carriers that determine the local photoconduction.

Action Taken 2.2 Figure R1 has been added to Supplementary Information as Fig. S6.

Question 2.3 Please include information on GPA analysis either in SI or in Methods.

Answer 2.3 GPA measurements were performed with an in-house developed software Strain++ (<http://jjppeters.github.io/Strainpp/>) using the 001 and 010 reference vectors and Gaussian masks with $\sigma = 0.87 \text{ nm}^{-1}$.

Action Taken 2.3 Information on GPA analysis is added to the Methods marked by red.

Response to Reviewer #3

Question 3.1 Even if the claims are correct, I doubt that it is worth for publication in Nature Communications which requires a significant breakthrough in the field. In my opinion, the study is just an extension of some of the previous works considering that the separation of light-induced electron-hole pair at the morphotropic phase boundary (MPB), mediated by flexoelectricity due to the huge strain gradient, is already well demonstrated from the work in Nat. Nanotech. 10, 972 (2015). Also, the direction of the strain gradient at both MPB in the mixed phase region is already reported in the work in Adv. Func. Mater. 25, 3405 (2015). Although the study contains some interesting observations, I do not recommend the paper for the publication in Nature

Communications.

Answer 3.1 We thank the Reviewer's detailed and constructive comments on our work. It can be perceived from the Reviewer's comments that our manuscript did not appropriately communicate the main message of our work to the Reviewer. Thus, we would like to highlight the novelty of our work and its importance to the optoelectronic and photovoltaic researches, especially in comparison with previous works.

Despite receiving increased attention these years, the potential effects of strain gradient on the photoelectric processes have been largely overlooked in the research of light sensors and solar cells, etc. In 2015, Chu *et al.* studied the spatially-resolved photocurrent distribution in the Ni/BiFeO₃/Pr_{0.5}Ca_{0.5}MnO₃/LaAlO₃ capacitor using a confocal microscopy with a laser spot of 450 nm in diameter, which show enhanced photocurrent at the morphotropic phase regions [Chu *et al. Nat. Nanotech.* **10**, 972(2015)]. However, this sub-micrometer resolution of the confocal microscopy can hardly reveal the detailed electronic features of the morphotropic phases with a dimension less than 100 nm. Although Chu *et al.* ascribed the photocurrent enhancement to the strain gradient, their explanation on how strain gradient is involved in the photoelectric process (which in our opinion is of fundamental importance) is rather speculative and elusive. For example, their model of built-in field across the phase boundary cannot explain the light polarization dependence of the photocurrent enhanced at the morphotropic phases (refer to **Answer 1.3**). This is partially due to the lack of information on local electronic properties, which is critical to understand local photoelectric processes.

On the contrary, our work directly characterized local electronic properties with nanometer lateral resolution by using the AFM system. With this detailed information, we demonstrated that there is a charge separation mechanism which happens in the morphotropic phase boundaries and it is based on the *flexo-photovoltaic* effect. As we have shown, the flexo-photovoltaic effect is due to the strain gradient-induced asymmetric distribution of light-excited carriers in the *k*-space, leading to charge separation in the real space [Yang *et al. Science* **360**, 904(2018)]. It is noteworthy that the flexo-photovoltaic effect is an intrinsic property originating from the strain gradient and it has nothing to do with depolarization field, polarization divergence, flexo-polarization or built-in field, but only with local symmetry breaking. This first demonstration of flexo-photovoltaic effect in the morphotropic phase boundary along with its important role played in local photoelectric properties could be considered as a significant breakthrough.

More importantly, this strain gradient mediated photoelectric process described in our work

applies universally to all the photo-active materials, offering new perspectives for understanding the structural-property relations in optoelectronic and photovoltaic devices regarding, e.g. grain boundaries, dislocation and interface, etc. Thus, we believe the novelty of our work is not lessened by previous work on strain gradients or strained BiFeO₃ films and thus can be published in Nature Communications.

Action Taken 3.1 The following sentence has been added to the manuscript in page 10 marked by red: “It is noteworthy that the flexo-photovoltaic effect is an intrinsic property originating from the strain gradient and it has nothing to do with depolarization field, polarization divergence, flexo-polarization or built-in field, but only with local symmetry breaking.”

***Question 3.2** It is important to mention what kind of tip was used (correspondingly the size of the tip radius) since the lateral dimension of the R/T' mixed region is ~100nm and it could be comparable or even smaller to the size of the radius for some AFM tips. It seems that height difference between R/T' boundary and T'/R boundary is less than 1nm which is much smaller than the reported value for BFO/LAO with a similar film thickness in previous papers (e.g. Science 326, 977 (2009), JAP 112, 064102 (2012)). This is usually the signature for using a tip with a large radius. If it is the case, the contact region between the tip and the sample could be not just below the center of the tip, and thus, it is difficult to determine the exact location of the current maxima or minima from the line profile in the manuscript.*

Answer 3.2 In this work, we used NSC14/Pt tips to perform the CFM characterization on strained BiFeO₃ films. As shown in **Figure R2**, the NSC14 tip possesses a diameter less than 25 nm, which is fine enough to characterize local electronic properties in strained BiFeO₃ films.

The small height difference (~1nm) shown in Figure 1c of the manuscript is due to local topography features, rather than caused by a scanning issue. The local height variation in the strained BiFeO₃/LaAlO₃ thin films is not only due to film thickness but also affected by local morphology. For example, the height difference between the peak and valley in the morphotropic phase region would reduce if local density of R/T' phases is high. As shown in **Figure R3**, a full

Figure R2. SEM image of the NSC14 tip used in our work. Image provided by the manufacturer

Figure R3. A full height profile extracted from Figure 1a of the manuscript.

line profile extracted from Figure 1a of the manuscript shows varied peak-to-valley height differences range from 0.6 nm to 2.5 nm depending on local structures. This also confirms our AFM scan resolution in both lateral and height directions is suitable to determine local topography and electronic properties.

Action Taken 3.2 The following sentence has been added to the Method part marked by red: “The NSC14/Pt AFM tip (Mikromasch) used in this work has a diameter less than 25 nm, which enables high lateral resolution in both topography and current mapping”

Question 3.3 *Authors claimed that R-phase matrix has a lower conductivity and tilted T' phase has a higher conductivity compared to one with T-phase region in the dark condition. If conductivity is only dependent on the type of structure and not influenced by conduction property*

in morphotropic phase boundary, why the currents on regions with tilted T' phase or with R phase are not uniform even in the regions away from the morphotropic phase boundary?

Answer 3.3 The absence of uniform plateaus in the current profile scanned over on R/T' morphotropic phases shown in Figure 1c of the manuscript is due to the local electronic properties. The CFM studies on strained BiFeO_3 films performed by other groups also shows similar saw-like features [see Kim *et al. NPG Asia Materials* **6**, e81(2014)]. More generally, this not only happens in the CFM measurement but also other types of scanning probe microscopy studies on the strained BiFeO_3 films, such as measurements of mechanical elasticity, stiffness, piezoresponse etc. [see Li *et al, Adv. Func. Mater.* **25**, 3405(2015); Sharma *et al, Adv. Mater. Interfaces* **3**, 1600033(2016); Cheng *et al. Sci Rep.* **5**, 8091(2015)].

As shown in **Figure R4**, the morphotropic R and T' -phases in the strained BiFeO_3 thin films combine with each other in an alternating/periodic way with a tilting angle. As both phases possess certain conduction, they will get involved in the current conduction to certain extent regardless of whether the AFM tip contacts R -phase or T' -phase. Specifically, when the CFM tip contacts the middle of the less conductive R -phase, the tip probes a low current (Figure R4a); when the tip moves towards the T' -phase, current would also flow in the conductive T' -phase after passing the less conductive R -phase, leading to an increased current magnitude (Figure R4b); when the tip contact directly at the middle of T' -phase, the dark current reach to its maximum value, forming thus the previously mentioned saw-like features. The plateaus in the scan profile would appear if one of the phases is fully insulating, thus blocking current injection from the CFM tip when the tip only contacts this phase. This occurs for example in the CFM scanning on the conductive domain walls embedded in the insulating domain matrix [see Volk *et al. Appl. Phys. Lett.* **110**, 132905(2017)]. Furthermore, even in the case that the phase boundaries is more conductive than the morphotropic phases, the CFM scan profile would also show the saw-like feature but with maxima appearing directly at the boundary locations. Clearly, this is not the case in our work. Thus, it is justified to conclude that the morphotropic T' -phase possesses an enhanced dark conduction compared to that of the R -phase.

Action Taken 3.3 Figure R4 along with associated comments has been added to Supplementary Information as Fig.S1.

Figure R4. Schematics shows the current flow under the conductive AFM tip while contacting (a) R-phase, (b) the boundary region and (c) T' -phase measured in the dark condition. The red arrow denotes the current conduction.

Question 3.4 Authors excluded the contribution of a bulk-photovoltaic effect considering its unidirectional nature by presenting schematic illustration in Fig. S3. However, I don't believe that the in-plane polarization in the mixed phase region should be unidirectional as schematically drawn in Fig. S3. The authors didn't present any in-plane PFM data and just assume the ideal unidirectional polarization distribution, meanwhile other works on the mixed-phase BFO show very complicated polarization distribution.

Answer 3.4 We thank the reviewer for this good suggestion to show in our work the PFM characterization on the morphotropic phase region, demonstrating the unidirectional polarization features (see **Figure R5**). In the out-of-plane direction, the strained BiFeO_3 film shows a uniform

Figure R5. Domain structure of the morphotropic region in strained BiFeO₃ thin film. (a) Topography, (b) out-of-plane PFM phase, (c) in-plane PFM phase, (d) Schematic of the ferroelectric polarization in the morphotropic *T* and *R* phases. PB denotes the morphotropic phase boundary. (e) Schematic showing the intrinsic bulk photovoltaic effect in the morphotropic *R* phase region. Red ball denotes the carriers delivered by the bulk photovoltaic effect of *R*-phase.

polarization direction pointing towards the substrate. In the in-plane direction, the polarization is aligned towards the right side of the scanned area. As schematically shown in Figure R5d, the “unidirectional polarization feature” refers to that each *R*-phase matrix in a morphotropic phase region possess the same polarization distribution and the *T'*-phase matrix also exhibits the same polarization features but different from that of *R*-phase, which is consistent with previous reports [Chen *et al. Adv. Mater.* **24**, 3070(2012); Chu *et al. Nat. Nanotech.* **10**, 972(2015)]. In addition to the simple domain patterns in the *R* and *T'* phase, domain configuration in the morphotropic phase boundaries is indeed very complicated probably due to its complicated structures connecting two types of morphotropic phases and strain gradient induced flexoelectric effect. Also, detailed characterization of the domain structures in the phase boundaries is challenging due to its small dimension and vulnerable structure which is very sensitive to external bias [Zhang *et al. Nat. Nanotech.* **6**, 98(2011)].

Overall, without taking the strain gradient-induced flexo-photovoltaic effect manifested at the phase boundaries into consideration, the intrinsic bulk photovoltaic effect manifested in the R -phase would not modulate the non-equilibrium carrier density and photoconduction of nearby T' -phase, as schematically shown in Figure R5e. Likewise, the bulk photovoltaic effect of the T' -phase would not alter the photoconduction of nearby R -phase.

Action Taken 3.4 Figure R5 is added into Supplementary Information as Fig. S5. The corresponding sentence in page 10 of the manuscript has been modified marked by red:

“Parenthetically, despite these morphotropic phases themselves being intrinsic non-centrosymmetric, their intrinsic bulk photovoltaic effect is unable to tailor local photoconductions due to the “unidirectional nature” of the ferroelectric polarization in each morphotropic phases.”

Question 3.5 *Since all the experimental data shown in this work are under the application of electric bias, the effect could be due to the combination of the photoconductivity and photovoltaic effects, yielding a difficulty for complete understanding of the observed phenomena.*

Answer 3.5 We thank the Reviewer for bringing the photovoltaic effect into discussion. In this work, the conductive AFM tip probes negligible current under illumination without applying external bias, as demonstrated in **Figure R6**. The photocurrent appears only if an external bias is applied. Thus, the current and its variation detected by the conductive tip under illumination are solely due to local photoconduction driven by external bias rather than photovoltaic or any combination effects.

Figure R6. Photocurrent mapping on strained BiFeO_3 thin film under 405 nm laser illumination with or without 2V external bias. (a) Scanned topography and (b) the corresponding current mapping. The time when external 2V bias is turned on or off is shown nearby the image. (c) Current profile on the scan area marked by red line in (b).

Action Taken 3.5 Figure R6 with above comment has been added to Supplementary Information as Fig. S4 and following sentence has been added to the manuscript at page 5 marked by red: “Note that the conductive tip probes negligible current without applying external bias under illumination (See Fig.S4 in SI).”

Question 3.6 (a) *What are the majority carrier and minority carrier?* (b) *The direction of the depolarization field due to the flexo-polarization should be toward R-phase, thus, electrons would move to the T' phase region and holes to the R phase region. In this sense, the majority carrier should be the hole. Is this correct?*

Answer 3.6 (a) Carrier type in the BiFeO₃ films is a fundamental but yet unresolved issue. Our strained BiFeO₃ thin films are grown in a PLD chamber by ablating a stoichiometric BiFeO₃ ceramic target. Due to bismuth volatility, the BiFeO₃ film is inherently bismuth deficient. This leads to the case of acceptor doping [Dedon *et al. Chem. Mater.* **28**, 5952(2016)]. For the sake of charge neutrality, these bismuth vacancies V_{Bi}'' are compensated by oxygen vacancies $V_O\ddot{}$ that act as electron donor. The final carrier type is determined by the competition between those four entities, i.e. electron, hole, V_{Bi}'' and $V_O\ddot{}$, which depends on bismuth and oxygen vacancy formation energies, the electronic energy level induced by V_{Bi}'' and $V_O\ddot{}$, the film growth temperature and oxygen pressure, etc. [see D. M. Smyth, “*The Defect Chemistry of Metal Oxides*”, Oxford University Press, Oxford (2000)]. Owing to its low conduction and high density of defects, determination of carrier type in the BiFeO₃ film is technically challenging, and is beyond the scope of the present work.

(b) As discussed in Answer 3.1, the charge separation mechanism happened in the morphotropic phase boundaries is proven to be the *flexo-photovoltaic* effect in our manuscript. This effect has nothing to do with depolarization field, flexo- polarization or built-in field [see the book “*The Photovoltaic and Photorefractive Effects in Noncentrosymmetric Materials*”; Young *et al.*, *Phys. Rev. Lett.* **109**, 16601(2012)]. Further information on flexo-photovoltaic effect and the related bulk photovoltaic effect please refers to our previous work and reference therein [e.g. Yang *et al. Appl. Phys. Lett.* **110**, 189302(2017)]. Based on our experimental observation, the flexo-photovoltaic effect would deliver holes to the R-phase and electrons into T'-phase if we assume holes are the majority carriers (see Figure R1). However, we cannot firmly state this without experimental evidence.

Action Taken 3.6 No action is taken here.

Question 3.7 *(a) The scale bar in Fig 1a is wrong. (b) The uppermost line for Fig 2c is not along with the ones for Fig 2a,b unlike the case in Fig 1. (c) “Fig. 3b” should be changed to “Fig. 2b” in line 116 of the manuscript*

Answer 3.7 We thank the reviewer for pointing for this. (a) The scale bar in Figure 1a has been corrected. (b) Figure 2c has aligned to Figure 2a,b. (c) “Fig. 3b” has been changed to “Fig. 2b” in line 116 of the manuscript.

Action Taken 3.7 Corrections have been done according to Reviewer’s comment.

Reviewers' comments:

Reviewer # 1 (Remarks to the Author):

Previous comments/response in blue

New comments in green

Question 1.2 A concern in the paper is that the authors do not take into account that the T' and R' phases have quite different band gaps (about 300 meV between them). If we consider the published absorption coefficient by Chen et al (APL 2010, 96, 131907), then at 3.06 eV (the light wavelength used here) the absorption coefficient of the T' phase is $\alpha_T \sim 1e5$ /cm; while for the R' phase it is $\alpha_R \sim 2.5e5$ /cm. This is a more than two times difference, which would imply that under illumination the various regions of the films would be absorbing light at different rates, which could possibly mean that the steady-state conditions (which is presumably how the films are measured by c-AFM) would involve different populations of carriers from the various phases. Could it be that the space charge field thus induced through this spatially-inhomogeneous generation of photo-charges could be the explanation for the observed phenomena? Although unlikely, the local variations in the photo-generated carriers, and thus different space charge fields generated, could be the reason for the observed results.

Answer 1.2 We thank the Reviewer for bringing the light absorption issue into discussion. Indeed, there is difference of a factor of two in the absorption coefficients of the strain-free rhombohedral BiFeO₃ grown on DyScO₃ substrate and the tetragonal phase grown in LaAlO₃ phase, according to Chen's work [APL 2010, 96, 131907]. However, it is necessary to clarify that the morphotropic R phase in the mixed phase region of the strained BiFeO₃/LaAlO₃ thin film is distinct from the typical rhombohedral BiFeO₃ phases. Although, for the sake of simplicity, we termed it in the manuscript as R -phase, the morphotropic R -phase is not the strain-free rhombohedral BiFeO₃ [Chen et al. Adv. Funct. Mater. 21, 133 (2011); Damodaran et al. Adv. Mater. 23, 3170 (2011); Chen et al. Phys. Rev. B 84, 094116(2011)]. Thus, it is not suitable to compare the bandgap and light absorption coefficient between the morphotropic phases in the strained BiFeO₃ films by using the values from the rhombohedral BiFeO₃/DyScO₃ thin film. An alternative way to get an insight into the bandgap values and light absorption coefficients of the morphotropic phases in the BiFeO₃/LaAlO₃ film is to study the thickness dependent absorption coefficient, as the volume of the R phase increases with the thickness. In this regard, our previous optical studies reveals a similar bandgap values and light absorption coefficients on a thin BiFeO₃/LaAlO₃ film, which consists of pure tetragonal-like T phase, and a thick BiFeO₃/LaAlO₃ film wherein the mixed morphotropic phases accounts for 50% area [Himcinschi et al. APL 106, 012908(2015)]. This indicates a negligible difference of bandgap ($\Delta E_g \leq 50$ meV) and absorption coefficient between the morphotropic phases in the strained BiFeO₃ films. As a results, the non-equilibrium carriers density induced by light should be similar in both R -phase and T -phase morphotropic phases of the strained BiFeO₃ films. Thus, the large contrast of the photoconduction between the phases cannot stem in the light absorption difference.

Action Taken 1.2 The sentence below has been added to the page 7 of the manuscript marked as red: "Also, given the negligible differences of bandgap and light absorption coefficient between the R - and T -phase, the incident light would generate similar density of non-equilibrium carriers in the morphotropic phases."

New question:

The authors claim that the band gap and therefore absorption properties of the R' like BFO is rather like that of the T' phase. In the following, I argue that indeed the band gaps are different, and thus ask the authors once again to consider that the phase-dependent optical absorption argument be considered and that they conduct a rigorous analysis to rule out a space-charge field effect.

Before beginning, I will define the nomenclature to avoid any confusion. T' BFO is the un-tilted T-like (monoclinic) phase of BFO (large axial ratio of about 1.25) when grown under strong compressive strain. T'_{tilt} is the tilted T' phase that appears in the mixed-phase striations. R' BFO is the almost completely relaxed (c = 3.97 Angstroms or so) phase that sometimes exists in mixed phase samples. S' BFO is the intermediate (strongly compressively strained R' BFO) that occurs in the mixed phase samples. Note that S' (without tilt) is usually not seen. S'_{tilt} – the phase that is sometimes incl. in this manuscript referred to as R'_{tilt} BFO – is in fact the strongly compressively strained R' BFO phase that appears in the mixed phase striations. In other words, I will describe the mixed phase regions as alternating zones of T'_{tilt} and S'_{tilt} phases.

In the authors previous response, they argue from Himcinschi et al. APL 106, 012908(2015) that the band gap of their T phase and mixed phase films is barely changed (this is given from the pure T-BFO film having a gap of 3.10 eV, while the mixed phase film has a slightly reduced gap of 3.05 eV). I would argue that this paper, while mostly correct, does not tell the full story.

The first issue is that the film considered to be a 'mixed' film had a thickness of about 118 nm. It is mentioned that the fraction of mixed phase in this film is about 50%. If that is the case, we can then surmise that in fact the S'_{tilt} phase comprises about 25% of the film, since the mixed phase itself is a 50/50 mix (approx.) of T'_{tilt} and S'_{tilt} phases.

To estimate the band gap of the mixed 118 nm thick film, we could consider the band gap to be some kind of 'linear combination' of the constituent phases (this is probably not strictly speaking true since the effective medium approximation is not necessarily applicable when the feature size starts to approach the light wavelength of several hundreds of nm). To do this, is important to know whether the S'_{tilt} phase has a band gap closer to the R' BFO phase or closer to T' BFO phase. For this, let us recall that the S'_{tilt} phase of BFO is under about 2.5 % of compressive strain (Z. Chen et al, Phys. Rev. B 84, 094116 (2011)). This is somewhat close to the level of misfit strain imparted by LSAT substrates (Infante et al PRL 2010).

There are a couple of works that consider the band gap of BFO under such levels of strain. One showed using ellipsometry that the band gap does increase slightly relative to the 'unstrained' R' BFO phase (on DSO for example). On LSAT (-2.5% strain, like the strain level seen in the R' tilt phase in mixed phase BFO films based on the lattice parameters), the band gap is about 2.81 eV (Sando et al. Nature Comms 7 10718 2016). Another study looked at BFO on NGO substrates (misfit approximately -2.7 %) – here they found a band gap of 2.82 eV (Liu et al. APL 103 181907 (2013)). Therefore, two independent measurements have shown a slightly larger band gap for R' BFO under ~2.5 % compressive strain.

To make a linear combination, we take 75 % of T-phase (3.1 eV) and 25 % of strained R-phase (2.81 eV) – yielding 3.0275 eV. This is not that different from the measurement of 3.05 eV in Himcinschi et al.

The second point to consider is concerning another important study reported by Dixit *et al.* in Advanced Science (150041, 2015). In that study the authors used EELs in their TEM measurements to show unequivocally that the S'_{tilt} phase *within* a mixed phase specimen does have a lower band gap than that of the T'_{tilt} phase. The difference in the band gaps is 0.35 eV, consistent with the ellipsometry measurements.

Therefore, based on the above arguments which demonstrate that the R'-like phase in their films should indeed have a different band gap, I request once again that the authors carry out further analysis. Namely, make some calculations that consider the different absorption cross sections of the different phases, calculate the density of charge carriers and then determine if the space charge field thus created could in fact explain their results.

Question 1.3 Regarding Figure 4, where there is a dependence on the light polarization, it is important to recall that the two BFO phases show quite a strong dependence of their absorption on the light polarization direction [see Schmidt et al., PRB 92, 075310 (2015)]. Also, we must remember that the symmetry of the T' phase is almost certainly MC, while the symmetry of the R' phase is probably monoclinic MA. Importantly these two phases will have their optical axes rotated by 45 degree relative to each other, in the plane – this would mean that changing the light polarization would probe various (and differing) proportions of the various complex refractive indices (and thus optical absorption) of the two phases. As mentioned above, although this scenario is unlikely; it would be appreciated that the authors to conduct some kind of analysis (using the papers given above as reference works) and justification that such a situation can be ruled out.

Answer 1.3 The non-centrosymmetric materials generally possess anisotropic light absorption depending on the light polarization direction with respect to the optical axis. As reported by Schmidt et al., the strained BiFeO₃/LaAlO₃ (001) thin film shows a uniaxial anisotropic absorption coefficients ($\alpha_a = \alpha_b \neq \alpha_c$) [Phys. Rev. B 92, 075310(2015)]. Accordingly, the light absorption coefficient of a linear polarized light in the strained BiFeO₃ film would remain a constant value independent of the light polarization direction if the light incidents along the out-of-plane direction (i.e. c direction). This is the strategy we used in our work to keep the light absorption unchanged while rotating the light polarization direction (see Figure 4a). Note that, as mentioned in Answer 1.2, the optical properties of the morphotropic R phase would be similar to that of the T' phase, rather than to the rhombohedral BiFeO₃ film grown in SrTiO₃ or DyScO₃ substrates. Also, the monoclinic distortion of the R phase ($0 \pm 0.3^\circ$) is even smaller than T' phase ($\sim 1.9^\circ$), indicating that the R phase is likely to possess the uniaxial optical properties ($\alpha_a = \alpha_b \neq \alpha_c$). Therefore, by illuminating the strained BiFeO₃ thin film along the out-of-plane direction with a help of the prime mirror, the light absorption in both morphotropic R and T' phases remain a constant value.

Action Taken 1.3 The following sentence has been added to the page 11 of the manuscript marked by red: “Due to the uniaxial optical properties of the strained BiFeO₃ thin films, i.e. the light absorption coefficient $\alpha_a = \alpha_b \neq \alpha_c$, the light absorption in the film would remain constant while rotating the light polarization angle.”

I appreciate the discussion; however, I think the authors misunderstood my meaning. First, while I understand that the light propagating along the z -axis ($//$ with [001]) means that absorption should in principle be not dependent on the light polarization, this is only true *if the light propagates in a direction parallel (i.e. along) the optic axis* (which is defined by the polarization direction of the ferroelectric, but has nothing to do with the monoclinic distortion of the unit cell, as mistakenly proposed above). Since both the T'_{tilt} and S'_{tilt} phases are monoclinic, their FE polarization direction, and therefore the optic axis, is tilted away from the [001] direction (this is even shown in the response figure R5). As a result, the complex refractive index (including absorption coefficient) is much more complicated and will display a dependence on the light polarization.

Consider the S'_{tilt} phase (the R' phase under about 2.5% compressive strain). The polarization (and therefore optic axis) are approximately along the [111] direction, if the film is unstrained. If we assume a simple pseudocubic cell, this implies that the angle between [001] and the polarization direction is 54.74 degrees.

Now, since the R' like phase in the mixed phase film is under about 2.5 % strain (as explained above), its polarization rotates slightly toward the out of plane [001] direction (see for instance Jang et al. PRL 101, 107602 – 2008 and Daumont et al. JPCM 24, 162202 – 2012). To estimate by how much the polarization rotates, we can use the polarization values from Daumont et al. as a rough guide. From that paper (the calculations part) at -2.5% strain, $P_z = 81$ uC/cm and total polarization is ~ 115 uC/cm². Therefore from trig we get an angle between [001] and P direction of $\text{asin}(81/115) = 44.8$ degrees. This means that the polarization rotates towards the out of plane direction, but it still not close to along the [001] direction.

Using the same arguments for the T' phase (i.e. the fact that it is monoclinic) and from the direct measurements of Schmidt et al [PRB 92, 075310 (2015)], the optic axis of T' like BFO is tilted by about 7.5 degrees from the [001] direction.

Now, if we consider the optical indicatrix for complex refractive index, we can decompose the response into the ordinary and extraordinary refractive indices (see for example ‘Fundamentals of Photonics’ by Saleh – Wiley 1991), as shown in the figure below:

Importantly, if you change the polarization of the incoming light, in one orientation e.g. P // [010] the light will be subject to the ‘ordinary’ absorption coefficient (‘ordinary ray’ in the Figure) so that

$$\tilde{n} = \tilde{n}_{T-o}.$$

On the other hand, if P // [100] the light will be subject to a combination of ‘ordinary’ and ‘extraordinary’ (‘extraordinary ray’ in the Figure) complex refractive indices, *i.e.*

$$\frac{1}{\tilde{n}_T^2(\theta)} = \frac{\cos^2 \theta}{\tilde{n}_{T-o}^2} + \frac{\sin^2 \theta}{\tilde{n}_{T-e}^2}$$

The same principle can be applied for the S'_{tilt} phase (R-like BFO phase, for which θ is ~ 45 degrees). (A cursory back of envelope calculation shows that while the contrast may be small between polarization 0 degrees and 90 degrees, each of the phases can have a different contrast relative to each other, and a rigorous treatment would be appropriate.)

Since this will bring about a contrast in the absorption coefficients for the different phases when the light polarization is changed, it could in principle explain the angular polarization dependence on their conductivity shown in Fig. 4(b).

I therefore request that the authors do a proper treatment with the known published optical constants (e.g. from Schmidt PRB and other words) to rule out the possibility that their results can be explained by the phase-dependent optical absorption, that also contains a dependence on the light polarization.

If the authors can convince that the above described effects are too weak to explain their results, their interpretation of the data with the polarization dependent ‘flexo-photovoltaic effect’ would be stronger. Until that, I believe there is enough doubt about the interpretation of the results that I would not recommend publication.

Reviewer #2 (Remarks to the Author):

The authors have addressed issues I have raised and I can now recommend the paper for publication.

Reviewer #3 (Remarks to the Author):

I think that the revised manuscript was quite improved. The authors tried to address most of questions, raised in the earlier review, by providing more experimental evidences and detailed explanations. They could definitely make this manucript better for readers. However, I still think that there remain some important issues which should be properly addressed before the publication in Nature Comm.

(1) They claimed, "a fingerprint feature of the flexo-photovoltaic effects is its dependence on the incident light polarization". The dependence is written as Eq. 1 in the manuscript. But the equation was orginally derived from the case of the strain gradients induced by a point force, such as due to the AFM tip or indentor. Is it still applicalble to the strain gradients developed in the phase boundary?

(2) In Eq. 1, they said that the theta is "the light polarization angle with respectve the current direction". In the experimental geometry, the current direction corresponded to the LaAlO₃ [010] direction. However, it might be more natural to think that the flexo-photovoltaic signal should depend on the angle with respectve to the strain gradient direction. I think that the authors need to do more careful explanations for readers.

(3) As I said in my last comment, there was a previous work which considered that the separation of light-induced electron-hole pair at MPB, mediated by flexoelectricity [Chu et al, Nature Nanotech. 10, 972 (2015)]. The major advancement of this work is that the authors could measure microscopic local electronic properties by enhancing the spacial resolution. Then, it would be important to derive how Eq. 1 will depend on the local electronic properties (R/T' conductance ratio).

(4) The authors claimed that their observed phenomena had nothing to do with the depolarization field, flexoelectric polarization, etc. I agree that the number of carriers generated could be depedent on the light polarization direction. However, as stated by Nat. Nanotech. 10, 972 (2015), the direction of the carrier movement could be influenced strongly by the depolarization field built by flexoelectric polarization at the phase boundary. When mean free path of the non-equilibrium carriers is comparable to the scale of strain-gradients, the carrier movement direction might be important.

(5) In the revised manuscript, the authors simply included new figures and data mostly in Supplementary Information. However, such simple inclusion makes the readers' job difficult. Since there are more space for Nature Comm., it would be much better to include many contents of Suppl. Information in the main text. In addition, many sentences are too lengthy and contain more than one key argument, so they are difficult to read.

Response to Reviewers' comments

“Strain-Gradient Mediated Local Conduction in Strained BiFeO₃ Film” by Ming-Min Yang *et al.*, Manuscript No. NCOMMS-18-34631R.

We would like to thank the reviewers for the thorough assessment of our work. Below, we have listed the issues/comments raised by the reviewers and our corresponding replies. We believe that we addressed all the issues raised by the Reviewers, thus making the message delivered by our manuscript clearer for readers.

Response to Reviewer #1

Question 1.1 Therefore, based on the above arguments which demonstrate that the *R*'-like phase in their films should indeed have a different band gap, I request once again that the authors carry out further analysis. Namely, make some calculations that consider the different absorption cross sections of the different phases, calculate the density of charge carriers and then determine if the space charge field thus created could in fact explain their results.

Answer 1.1 We thank the Reviewer for the detailed elucidation of the question. Now we agree with the Reviewer that the *R*-phase in the mixed phase region (i.e. the S'_{tilt} phase referred by the Reviewer) might have a lower bandgap than the nearby morphotropic *T*'-phase (named as T'_{tilt} by the Reviewer). However, the associated light absorption contrast between *R*-phase and the *T*'-phase cannot explain the measured photoconduction contrast showed in the Manuscript.

To analysis the possible role of the light absorption contrast, as suggested by the Reviewer, we assume the light absorption coefficient of the *R*-phase similar to the rhombohedral BiFeO₃ phase grown in DyScO₃ substrate ($\alpha_R = 2.5 \times 10^5 \text{ cm}^{-1}$ for 405 nm light). The light absorption coefficient of the matrix *T*-phase and the morphotropic *T*'-phase under the illumination of 405 nm laser equals to $\alpha_T = 1.23 \times 10^5 \text{ cm}^{-1}$ [*Appl. Phys. Lett.* **96**, 131907 (2010)]. Given the film thickness as 100 nm, the *R*-phase would absorb 92% of the light while 8% of the light is transmitted. In the case of the matrix *T*-phase and the morphotropic *T*'-phase, about 70% of the light is absorbed with 30% light transmitted. Note that the light reflection by the film surface is assumed to be homogenous over different phases and is not considered here. Thus, the *R*-phase absorbs 1.3 times of light than that of matrix *T*-phase and the morphotropic *T*'-phase under the 405 nm laser illumination. Provided that the density of photo-excited carriers is proportional to the absorbed light intensity, more (about 30%) non-equilibrium carriers are excited in the *R* -phase

Fig.R1. Inhomogeneous light absorption related diffusion process. (a) Schematic shows the variation of the light-induced nonequilibrium carrier density in the morphotropic phase region due to the stronger light absorption in the *R*-phase under 405 nm laser illumination. Here, the diffusion process is not yet considered. (b) Diffusion induced non-equilibrium carrier redistribution and associated space charge field. Here, the photo-excited carriers are assumed to be holes, resulting in a space charge field pointing from *T'*-phase to the *R*-phase.

than those in the matrix *T*- and the morphotropic *T'*-phase. Note that this light-absorption variation itself would not induced space charge field as each part of the phases is still in neutral charge (see Fig. R1a). Due to the higher density of non-equilibrium carriers in the *R*-phases, non-equilibrium carriers diffuse from the *R*-phase to the *T'*-phases, resulting in space charge of which field prevents further carrier diffusion (see Fig. R1b). Consequently, the density of non-equilibrium carriers in the *T'*-phase is increased, resulting in

$$n_{R-phase} > n_{T'-phase} > n_{T-phase}$$

where n is the density of non-equilibrium carriers. As the conduction is proportional to the density of carriers, this would give rise to the photoconduction contrast as

$$\sigma_{R-phase} > \sigma_{T'-phase} > \sigma_{T-phase}$$

Clearly, this is inconsistent with experimental observation presented in the manuscript wherein $\sigma_{R-phase} > \sigma_{T-phase} > \sigma_{T'-phase}$. Thus, the light absorption contrast between morphotropic phases cannot account for the experimentally observed photoconduction contrast.

Fig. R2. Spatially resolved photocurrent distribution under 365 nm light illumination. (a) Surface topography and (b) photocurrent distribution characterized under illumination of 365 nm light. (c) Profile comparison between the photocurrent and surface morphology of the area marked by blue arrow in (a). The light intensity is about 200 mW/cm² and a bias of 3 V is applied to the side Pt electrode.

To further demonstrate the crucial role of the flexo-photovoltaic effect and especially, to exclude the effect of light absorption variation, we performed further experiments and mapped the spatial distribution of the photoconduction over BiFeO₃/LaAlO₃ thin film under the illumination of 365 nm light. The corresponding photon energy is well above BiFeO₃ band gap energy ($h\nu = 3.41 \text{ eV} \gg E_g^{BFO}$). According to Chen *et al.*, the less-strained rhombohedral BiFeO₃ and the strained BiFeO₃/LaAlO₃ thin films exhibit similar absorption coefficient at 365 nm ($\alpha \approx 2.8 \times 10^5 \text{ cm}^{-1}$) [*Appl. Phys. Lett.* **96**, 131907 (2010)]. Thus, it would be reasonable to assume a homogenous light absorption over the morphotropic phases in the strained BiFeO₃ films. As shown in Fig.R2, there exists clear photoconduction contrast over the scanned area consisting of matrix *T*-phase, morphotropic *T'*-phase and *R*-phase. As profiled in Fig. R2c, the current probed at the *R*-phase is about three times larger than that collected at the nearby *T'*-phase. Therefore, the light absorption variation in the strained BiFeO₃ thin film do not play a major role in tuning local photoconduction.

Action Taken 1.1 Discussion about the effect of light absorption inhomogeneous in strained BiFeO₃ thin film has added to Supplementary Information along with Fig. R1 and Fig. R2. Following sentence has added to page 7 of manuscript marked as red:

“Also, this photoconduction contrast cannot be simply attributed to the optical absorption difference between morphotropic phases, as detailed in the Note S1 of SI.”

Question 1.2 I therefore request that the authors do a proper treatment with the known published optical constants (e.g. from Schmidt PRB and other words) to rule out the possibility that their results can be explained by the phase-dependent optical absorption, that also contains a dependence on the light polarization.

Answer 1.2 We would like to thank the Reviewer again for the detailed explanation of the question. Owing to the tilting of the ferroelectric polarization away from the surface normal direction (i.e. $[001]_{pc}$), both morphotropic phases (namely, R -phase and T' -phase) in principle possess certain light polarization dependent absorption. Although we have confirmed in Answer 1.1 that the light absorption difference between morphotropic phases do not account for the observed photoconduction contrast, the potential impact of its light-polarization dependence on local photoelectric properties deserves further exploration. To this end, we measured the light polarization dependent photoconduction of $\text{BiFeO}_3/\text{LaAlO}_3$ (001) thin film with normal illumination in respect to the surface. Here, the $\text{BiFeO}_3/\text{LaAlO}_3$ (001) film consists solely of monoclinic T -phase with a thickness of 40 nm and the coplanar electrodes are perpendicular to the in-plane net ferroelectric polarization (see Fig. R3a). As shown in Fig. R3b, the in-plane photoconduction of the T -phase BiFeO_3 stays almost constant while rotating the incident light polarization, probably due to the small mismatch angle between its optical axis and surface normal direction in $\text{BiFeO}_3/\text{LaAlO}_3$ films. Assuming no dependence of carrier mobility on the light polarization, the above result points to negligible light polarization dependence of the photo-excited carrier density in T -phase BiFeO_3 . Due to the highly similarity between the matrix T -phase and the tilted T' -phase in the mixed phase region, it would be reasonable to claim the negligible dependence of the photoelectric properties in the T' -phase on light polarization illuminating along surface normal direction. This is also consistent with previous report using out-of-plane capacitor geometry [K. Chu *et al.*, *Nat. Nanotech.* **10**, 972(2015)].

In the case of morphotropic R -phase in the mixed region, it is difficult to directly probe the intrinsic light polarization dependence of its photoelectric properties due to its small dimension and strong interaction with its surroundings. To circumvent this difficulty, we studied the less strained “rhombohedral” $\text{BiFeO}_3/\text{SrTiO}_3$ (001) film consisting of a mono-domain structure. Details about this sample can refer to our previous work [M.M. Yang *et al.*, *Appl. Phys. Lett.* **110**, 183902(2017)]. Fig. R3c illustrates the measurement configuration wherein the in-plane electrodes run perpendicular to the in-plane ferroelectric polarization and light incidents along the surface normal direction. As shown in Fig. R4d, the photoconduction of the $\text{BiFeO}_3/\text{SrTiO}_3$ (001) film exhibits strong light polarization dependence. The photoconduction reaches its maxima while

Fig.R3. Effect of light polarization dependent optical absorption on the BiFeO₃ with different structures. (a) Schematic shows the measurement geometry and (b) the corresponding light polarization dependence of the photoconduction in BiFeO₃/LaAlO₃ thin films consisting of *T*-phase. (c) Schematic shows the measurement geometry and (d) the corresponding light polarization dependence of the photoconduction in BiFeO₃/SrTiO₃ (001) thin films consisting of a single ferroelectric domain. In (a)-(d), the light polarization angle refers to that made between light polarization and the in-plane net polarization direction. (e) Schematic illustrates the light polarization dependence of the photoconduction ratio between *R* and *T'*-phases if it is the light polarization dependence of the optical absorption that determines local photoelectric properties while tailoring the light polarization angle.

light polarization running parallel to the in-plane ferroelectric polarization; whereas, it reduces to its minima when light polarization is perpendicular to the ferroelectric polarization. Note that the ferroelectric polarization of morphotropic *R*-phase in the strained BiFeO₃/LaAlO₃ films also tilts towards the $[111]_{pc}$ direction as in the case of the “rhombohedral” BiFeO₃/SrTiO₃ films. It is likely that the *R*-phase shows the similar dependence of photoelectric properties on light polarization with smaller amplitude in variation. Therefore, the *R*-phase would possess the maximum photoconduction when light polarization runs along its in-plane ferroelectric polarization.

If it is the light polarization dependence of optical absorption that determines the local photoconduction variation while rotating the light polarization (see Fig.4 of the Manuscript), the photoconduction contrast between the *R*-phase and the *T'*-phase should maximize when light

polarization is parallel to the in-plane ferroelectric polarization of R -phase (i.e. $\theta = 45^\circ$). Afterwards, it reaches a minimum value while light polarization equals to $\theta = 135^\circ$, as schematically illustrated in Fig. R3e. Apparently, this is inconsistent with experimental result shown in Fig. 4 of the Manuscript. Therefore, we can conclude that the polarization dependent optical absorption do not play a major role in the local photoconduction. Instead, as demonstrated in the Manuscript, it is the flexo-photovoltaic manifested at the morphotropic phase boundaries that controls local photoelectric properties in the mixed phase region.

Action Taken 1.2 Fig. R3 along with above discussion has been added to the Note S1 Supplementary Information. The following sentence has been added to the page 12 of Manuscript marked as red:

“Despite that the morphotropic phases might have a certain light polarization dependent optical absorption due to the tilting of ferroelectric polarization away from surface normal direction, their light polarization dependent photoconduction cannot account for the sinusoidal dependence shown in Fig. 4c (see Note S2 in SI).”

Response to Reviewer #3

Question 3.1 They claimed, “a fingerprint feature of the flexo-photovoltaic effects is its dependence on the incident light polarization”. The dependence is written as Eq. 1 in the manuscript. But the equation was originally derived from the case of the strain gradients induced by a point force, such as due to the AFM tip or indenter. Is it still applicable to the strain gradients developed in the phase boundary?

Answer 3.1 First of all, we appreciate the positive comment from the Reviewer on our manuscript revision. Although Eq. 1 in the manuscript originally derived from the case of a point force induced strain gradient, it is still applicable to express the light polarization dependence of the flexo-photovoltaic effect manifested at the morphotropic phase boundaries owing to the same origin. Nevertheless, it is a good suggestion to derive a formula specifically for the case of the flexo-photovoltaic effect in morphotropic phase boundaries.

We assume that the flexo-photovoltaic effect in the morphotropic phase boundaries can be effectively expressed by a third-rank tensor β_{ijk}^{eff} :

$$\beta_{ijk}^{eff} = \begin{pmatrix} \beta_{11} & \beta_{12} & \beta_{13} & \beta_{14} & \beta_{15} & \beta_{16} \\ \beta_{21} & \beta_{22} & \beta_{23} & \beta_{24} & \beta_{25} & \beta_{26} \\ \beta_{31} & \beta_{32} & \beta_{33} & \beta_{34} & \beta_{35} & \beta_{36} \end{pmatrix} \quad \text{R1}$$

The incident light along the surface normal direction (i.e. z direction) can be expressed as:

$$e = (\cos \theta, \sin \theta, 0) \quad \text{R2}$$

where θ is the angle between the light polarization and LaAlO_3 $[010]_{pc}$ direction. Note that the coordinate x is set parallel to the LaAlO_3 $[010]_{pc}$ direction and y is parallel to the $[100]_{pc}$ direction. Then, the light polarization field $e_j e_k$ equals to

$$e_j e_k = (\cos^2 \theta, \sin^2 \theta, 0, 0, 0, 2 \sin \theta \cos \theta)^T \quad \text{R3}$$

In the case of the morphotropic phase boundary (see Fig. 4c in the Manuscript), the flexo-photovoltaic current perpendicular to the interface, namely, the current along the $[100]_{pc}$ direction, plays the major role in mediating local photoconduction. Thus, the flexo-photovoltaic current I_{FPV} can be given as:

$$\begin{aligned} I_{FPV} &= I_0 A \beta_{2jk}^{eff} e_j e_k = I_0 A (\beta_{21} \quad \beta_{22} \quad \beta_{23} \quad \beta_{24} \quad \beta_{25} \quad \beta_{26}) \begin{pmatrix} \cos^2 \theta \\ \sin^2 \theta \\ 0 \\ 0 \\ 0 \\ 2 \sin \theta \cos \theta \end{pmatrix} \\ &= I_0 A (\beta_{21} \cos^2 \theta + \beta_{22} \sin^2 \theta + 2 \beta_{26} \sin \theta \cos \theta) = I_0 A [B_1 + B_2 \cos(2\theta + \varphi)] \end{aligned} \quad \text{R4}$$

where $B_1 = \frac{1}{2} \beta_{21} + \frac{1}{2} \beta_{22}$, $B_2 = \sqrt{\frac{1}{4} (\beta_{21} - \beta_{22})^2 + \beta_{26}^2}$, φ is the offset angle, I_0 is the light intensity, A is the morphotropic phase boundary area. Equation R4 is similar to the Equation 1 used in the manuscript. The offset angle φ is fitted to be 6° in the Fig. 4 of the Manuscript.

Action Taken 3.1 Above calculation is added to the Method Section of the manuscript and Equation 1 in the manuscript is substituted by Eq. R4.

Question 3.2 In Eq. 1, they said that the θ is “the light polarization angle with respect to the current direction”. In the experimental geometry, the current direction corresponded to the LaAlO_3 $[010]$ direction. However, it might be more natural to think that the flexo-photovoltaic signal should depend on the angle with respect to the strain gradient direction. I think that the authors need to do more careful explanations for readers.

Answer 3.2 The angle θ is the angle made between light polarization and the LaAlO_3 $[010]_{pc}$, as illustrated in the inset of Fig. 4c. The morphotropic phase boundaries at the surface of the $\text{BiFeO}_3/\text{LaAlO}_3$ film are either parallel to the LaAlO_3 $[100]_{pc}$ or $[010]_{pc}$. Thus, the direction of the strain gradient are also either parallel to the LaAlO_3 $[010]_{pc}$ or $[100]_{pc}$ directions. To make the definition of the angle clearer for the reader, corresponding modification has been made.

Action Taken 3.2 The sentence “ θ is the light polarization angle with respect to the current direction.” has been replaced by the “ θ is the angle made between light polarization and the LaAlO_3 $[010]_{pc}$ direction (see the inset of Fig. 4c)”.

Question 3.3 As I said in my last comment, there was a previous work which considered that the separation of light-induced electron-hole pair at MPB, mediated by flexoelectricity [Chu et al, Nature Nanotech. 10, 972 (2015)]. The major advancement of this work is that the authors could measure microscopic local electronic properties by enhancing the spatial resolution. Then, it would be important to derive how Eq. 1 will depend on the local electronic properties (R/T conductance ratio).

Answer 3.3 The Eq. 1 of the manuscript is typical for bulk photovoltaic effect. It relates the photogenerated current with the light polarization direction relative to the crystallographic structure of the sample. In the normal bulk photovoltaic effect, the Eq. 1 is only determined by the bulk photovoltaic tensor β_{ij} , which depends solely on the symmetry of the crystal. We have shown in the Methods and Answer 3.1 how can we estimate this relationship, if we will know exactly the β_{ij} tensor components. This means the functional relation between the flexo-photovoltaic current and the light polarization, which is formulized in Eq.1, are only determined by the local symmetry and structure of the morphotropic phase boundary in strained BiFeO_3 films. The local electronic properties of the boundaries, such as light absorption coefficient, carrier mobility, effective carrier mass etc., will only affect to some extent the *magnitude* of the flexo-photovoltaic current. However, it will not affect the functional dependence on the light polarization which, as we reiterate, is only determined by the local symmetry breaking.

To study the intricate correlation between the flexo-photovoltaic constants β_{ij} , local electronic properties and the strain gradient will require a multi-annual research program even for a simple homogenous material. This is clearly out of the scope of present work.

It is worth noting that in a “classical” photovoltaic effect based on variation of *electronic* properties (such as Fermi level or band structure) without any local symmetry breaking, the photocurrent is generated only by drift-diffusion of thermalized electrons. There will be no dependency on the light polarization due to isotropy of the carriers in the conduction band.

Furthermore, the electronic properties of the R - and T' -phase certainly will not affect the flexo-photovoltaic current generated in the phase boundary. Instead, the photoelectric properties of these morphotropic phases are tailored by the flexo-photovoltaic effect of the boundaries, which is the main topic of present Manuscript.

We believe that our work is experimentally demonstrating the critical role of the flexo-photovoltaic effect in the photoelectric properties. This was largely concealed until now. As confirmed by Fig.4 of the Manuscript, the flexo-photovoltaic current maximizes the photoconduction ratio between R - and T' -phase when light polarization is parallel to the phase boundaries. When light polarization is perpendicular to the phase boundary, the flexo-photovoltaic current minimizes, resulting the least photoconduction contrast between R - and T' -phase.

Action Taken 3.3 The following sentence has been added to Page 12 of the manuscript marked by red.

“Also, this indicates the flexo-photovoltaic effect in the morphotropic phase boundary maximizes when light polarization is parallel to the phase boundary whereas minimizes when light polarization is perpendicular to the boundary.”

Question 3.4 *The authors claimed that their observed phenomena had nothing to do with the depolarization field, flexoelectric polarization, etc. I agree that the number of carriers generated could be dependent on the light polarization direction. However, as stated by Nat. Nanotech. 10, 972 (2015), the direction of the carrier movement could be influenced strongly by the depolarization field built by flexoelectric polarization at the phase boundary. When mean free path of the non-equilibrium carriers is comparable to the scale of strain-gradients, the carrier movement direction might be important.*

Answer 3.4 We believe that broken inversion symmetry is the unique origin of all these effects, including polarization and associated effects, bulk photovoltaic effect, flexoelectric effect, etc. The flexo-photovoltaic effect is an intrinsic property of a material of which symmetry is broken

by the strain gradient. Similar to the bulk photovoltaic effect, the flexo-photovoltaic effect is not due to the flexoelectric polarization nor the depolarization field.

We agree that the strain gradient not only induces the flexo-photovoltaic effect but also the flexoelectric polarization in the morphotropic phase boundary. The flexoelectric polarization would induce a depolarization field only if the polarization is not fully compensated and polarization divergence $-\nabla \cdot \vec{P}$ appears. In this case, the photo-excited non-equilibrium carriers at the morphotropic phase boundary would not only subject to the flexo-photovoltaic effect but also the depolarization field. However, the appearance of the depolarization field at morphotropic phase boundary is speculative. The paper “*Nat. Nanotech.* **10**, 972 (2015)” is just stating without a clear-cut experimental evidence that the hypothetical depolarization field plays a role in charge separation. The spatial distribution of the ferroelectric polarization and the corresponding polarization divergence in the morphotropic phase region is inferred from the assumption that electrical polarization is linearly proportional to the piezoresponse vector. This naive assumption (commented by the authors themselves) clearly is not applicable to the morphotropic phase boundaries of which large piezoresponse is due to the subtle and intricate transformation between those morphotropic phases each with its own symmetry and piezoelectric tensor, rather than a simple proportionality on the magnitude of the ferroelectric polarization [J. Zhang *et al.*, *Nat. Nanotech.* **6**, 98 (2011)]. At a more detailed analysis, the derived distribution of the polarization divergence $-\nabla \cdot \vec{P}$ and the associated depolarization field in the morphotropic phase region, as the result of this naive assumption, is inconsistent with the symmetry of the strain gradient distribution. As shown in Fig. 3 of our manuscript, the strain gradient shows a mirror-type symmetrical distribution in morphotropic phase region. Specifically, strain gradient $\partial \varepsilon_{yy} / \partial l$ at the boundaries located at the right and left side of the T' -phase is always pointing to the T' -phase (see Fig. R4a). Thus, the resultant flexoelectric polarization and associated depolarization field, if any, would also exhibit similar mirror like symmetry, namely pointing to the opposite direction with respect to the T' -phase (see Fig. R4b,c) [also see Y.-J. Li *et al.*, *Adv. Funct. Mater.* **25**, 3405 (2015)]. By the

Fig. R4. Symmetry of the strain gradient distribution and associated effects. Directions of (a) strain gradient, (b) polarization induced by flexoelectric effect and (c) associated depolarization field at nearby morphotropic phase boundaries enclosing the T' -phase. (d) The depolarization field directions derived in the paper “*Nat. Nanotech.* **10**, 972 (2015)”.

contrary, in the paper “*Nat. Nanotech.* **10**, 972 (2015)”, the depolarization field at the nearby phase boundaries is assumed always to be in the same direction (see Fig. R4d).

Moreover, if the depolarization field exists at the morphotropic phase boundaries due to the large polarization divergence $-\nabla \cdot \vec{P}$, this polarization divergence would also induce charge accumulation at the phase boundaries, leading to an enhanced conduction at the boundaries, as in the case of ferroelectric domain walls [see J. Seidel *et al.*, *Nat. Mater.* **8**, 229(2009); A. Crassous *et al.*, *Nat. Nanotech.* **10**, 614 (2015)]. However, the dark cAFM characterization shown in Fig. 1 of our manuscript did not show any conduction enhancement at phase boundaries. Therefore, it would be reasonable to claim that the polarization divergence and depolarization field is minimal at the morphotropic phase boundary. Thus, the existence and the manifestation of the depolarization field at the morphotropic phase boundaries is highly questionable and speculative and do not play a major role in mediating local photoelectric properties.

Since our paper claims and also experimentally demonstrates another origin of the effect, we prefer to let the reader to draw the appropriate conclusion and potentially perform further investigation.

Action Taken 3.4 We prefer not to include any above comments in our manuscript. Thus, no action is taken.

Question 3.5 In the revised manuscript, the authors simply included new figures and data mostly in Supplementary Information. However, such simple inclusion makes the readers' job difficult. Since there are more space for Nature Comm., it would be much better to include many contents of Suppl. Information in the main text. In addition, many sentences are too lengthy and contain more than one key argument, so they are difficult to read.

Answer 3.5 This is a good suggestion, but in the other side too many technical details might break the reading fluency. Fig. S6 has been introduced into the manuscript as part (a) of Fig. 4. The analysis and calculation following Fig. S6 in the Supplementary Information has been moved to the Methods part of the Manuscript. Additionally, some long sentences have been revised.

Reviewers' comments:

Reviewer #1 (Remarks to the Author):

The authors have included extra discussion and have carried out further experiments to address the questions of the Reviewers. The extra experiments with 365 nm light excitation are a welcome addition; it appears to give good evidence that light absorption and generation of local SC fields is not playing a significant role.

One point that I find confusing however is that although the authors have attempted to be convincing in their reply to Reviewer #3, they have not included any further information/explanation in the Supplementary Information (SI). On the other hand, for the questions from Reviewer #1 they have included their explanation in the SI. I feel that if they are sure of their explanation and are convinced of its accuracy, the authors should not be afraid to include such details in the SI, as surely readers are likely to have same questions as the Reviewers. Therefore, I strongly suggest that all the detailed discussions be included in the SI. In general, while I am still slightly unconvinced by the origin of their results (there may be still enough ambiguity), the authors presented good set of convincing data, and the additional supplementary notes (after considering the above point regarding Reviewer #3's comments) are sufficient that the readers can make up their own minds. With this in mind, I believe the paper is suitable for publication.

A final note: the title in its current form appears grammatically incorrect (or at least clumsy). It should either read "Strain-Gradient Mediated Local Conduction in a Strained BiFeO₃ Film" or "Strain-Gradient Mediated Local Conduction in Strained BiFeO₃ Films". Presumably these results are on one film only so probably the former is more accurate.

Reviewer #3 (Remarks to the Author):

The authors made detailed replies to my previous comments. Accordingly, they clarified most of the questions and misunderstandings from their previous submission. And they improved the discussion and explanation of concept of their works. Particularly, I am pleased that they have derived the flexo-photovoltaic effect specifically for the case of morphotropic phase boundary at mixed phase BFO.

I think that the paper is almost ready for publication. However, I still have one minor (but could be important) concern related to the Figure 4, which shows that the conductance ratio between R/T phase show light polarization dependent photoconduction. According to the schematic diagram in Fig 4, the conductance ratio of R/T phase looks like to be related to the $(n_{ph,R} + n_{fpv}) / (n_{ph,T} - n_{fpv})$, where n denotes the light generated non-equilibrium carrier density. Considering the scenario assumed in the manuscript, isn't it natural to think that conductance at R phase or T phase should present polarization dependent photoconduction rather than the ratio between them? For the general readers, the authors had better to explain clearly how their proposed mechanism in the current manuscript could explain the light polarization dependency of R/T phase conductance ratio.

Response to Reviewers' comments

“Strain-Gradient Mediated Local Conduction in Strained BiFeO₃ Films” by Ming-Min Yang *et al.*, Manuscript No. NCOMMS-18-34631B.

We would like to thank the reviewers for the thorough assessment of our work. Below, we have listed the issues/comments raised by the reviewers and our corresponding replies. We believe that we addressed all the issues raised by the Reviewers, thus making the message delivered by our manuscript clearer for readers.

Response to Reviewer #1

***Question 1.1** One point that I find confusing however is that although the authors have attempted to be convincing in their reply to Reviewer #3, they have not included any further information/explanation in the Supplementary Information (SI). On the other hand, for the questions from Reviewer #1 they have included their explanation in the SI. I feel that if they are sure of their explanation and are convinced of its accuracy, the authors should not be afraid to include such details in the SI, as surely readers are likely to have same questions as the Reviewers. Therefore, I strongly suggest that all the detailed discussions be included in the SI.*

A final note: the title in its current form appears grammatically incorrect (or at least clumsy). It should either read “Strain-Gradient Mediated Local Conduction in a Strained BiFeO₃ Film” or “Strain-Gradient Mediated Local Conduction in Strained BiFeO₃ Films”. Presumably these results are on one film only so probably the former is more accurate.

Answer 1.1 We appreciate the Reviewer’s recommendation to publish our paper. Per the Reviewer’s suggestion, we have included the discussion with Reviewer #3 about the depolarization field into the Supplementary Information. Regarding the manuscript title, we have modified it to “Strain-Gradient Mediated Local Conduction in Strained BiFeO₃ Films” as this effect has been observed repeatedly on many strained BiFeO₃ thin films.

Action Taken 1.1 Discussion with Reviewer #3 about the role of depolarization field induced by flexoelectric effect has been added to the Supplementary Information as Note S1. Following sentence has added to page 10 of manuscript (marked in red):

“Also, the depolarization field derived from the flexoelectric polarization would play a minima role here, if any (see Supplementary Note S1).”

Response to Reviewer #3

Question 3.1 *I still have one minor (but could be important) concern related to the Figure 4, which shows that the conductance ratio between R/T phase show light polarization dependent photoconduction. According to the schematic diagram in Fig 4, the conductance ratio of R/T phase looks like to be related to the $(n_{ph,R}+n_{fpv})/(n_{ph,T}+n_{fpv})$, where n denotes the light generated non-equilibrium carrier density. Considering the scenario assumed in the manuscript, isn't it natural to think that conductance at R phase or T phase should present polarization dependent photoconduction rather than the ratio between them? For the general readers, the authors had better to explain clearly how their proposed mechanism in the current manuscript could explain the light polarization dependency of R/T phase conductance ratio.*

Answer 3.1 We would like to thank the Reviewer for bring the light polarization dependence into further discussion. Indeed, the photoconduction of the R - and T' -phase in the morphotropic phase boundaries depend on the incident light polarization. To demonstrate this dependence, we characterized the respective photoconduction of R - and T' -phase by scanning method. Specifically, we mapped the spatially resolved photocurrent distribution of the same area containing morphotropic R - and T' -phase and matrix T -phase under illumination with various light polarization direction. To mitigate measurement errors and uncertainties, the photocurrent of each phases at a certain light polarization direction is obtained by averaging the photocurrent acquired in scanned area. Note that, apart from the light polarization direction, the magnitude of the photocurrent acquired during scanning processes is also vulnerable to other external factors, such as the electrical contact quality between the conductive tip and sample surface. To make our measurements more robust, we exploited a photoelectric feature of the matrix T -phase, i.e., the independence of its photoconduction on the light polarization (see Fig. S10 in Supplementary Information). Accordingly, we use the photocurrent obtained on the matrix T -phase as the baseline for each scan and normalize the photocurrent of R - and T' -phase to that of matrix T -phase. This mitigates the extrinsic variations and reveals an intrinsic correlation of local photoelectric properties on the light polarization direction (see Fig. R1). We prefer this dynamic measurement method to the fixed-point method as the latter would induce substantial error due to the thermal drift, small dimension of those morphotropic phase, etc.

As shown in Fig. R1a, the photoconduction ratio between morphotropic R -phase and the matrix T -phase shows sinusoidal dependence on the light polarization, which can be well fitted by Eq. 1 of the manuscript. The photoconduction ratio between T' and matrix T -phase also depends on

the light polarization but with a 90° phase shift compared to that of R/T conductance ratio. This is also consistent with our proposed model. Note that the variation amplitude of the T'/T photocurrent ratio is much smaller than that of the R/T ratio. This is probably due to the entangled conduction path underneath the AFM tip. As illustrated in the insert of Fig. R1b, the nearby R -phase with enhanced conductance would also contribute to the conduction when the tip contacts the T' -phase with depressed photoconduction. The enhanced photoconduction of the R -phase would compensate somehow the reduced photoconduction of the T' -phase while rotating the light polarization, resulting in a small variation amplitude. Due to the small variation in the amplitude of the T'/T conductance ratio, the light polarization dependent conductance ratio between R/T' shown in the Fig.4 of the manuscript can also be well fitted with Eq.1. Since the main topic of present manuscript focuses on the strain gradient mediation of the R - and T' -phase photoconduction, we prefer to show the photoconductance ratio between the R - and T' -phase in the main text.

Fig. R1. Light polarization dependent photoconduction ratio between (a) R -phase and T -phase; (b) T' -phase and T -phase. The solid curve is the fit of the experimental data with Eq. 1.

Action Taken 3.1 The characterization details about the light-polarization dependent photoconduction mapping has been added to the Method section of the manuscript. Fig. R1 along with above discussion has been included in the Supplementary Information. Following sentence has been added to page 12 of the manuscript:

“Also, the light polarization dependence of the photo-conductance ratio between the morphotropic T' -, R -phase and matrix T -phase is given in Supplementary Fig. S6.”

REVIEWERS' COMMENTS:

Reviewer #3 (Remarks to the Author):

I think that the authors made the detailed reply to my previous question and included the previous discussion related to the depolarization field at the phase boundary. The reason why the conductance ratio of R/T' phase present the cosinusoidal behavior is that the amplitude of the T'/T photocurrent ratio is much smaller than that of R/T ratio. To make the paper more solid, I suggest authors to change the figure 4 and/or clearly mention the reason for the cosinusoidal behavior in the conductance ratio of R/T' phase. I will prevent any confusion of the readers. However, such change can be easilly addressed. I have no further objection for the publication of this paper in Nature Communication.

Response to Reviewers' comments

“Strain-Gradient Mediated Local Conduction in Strained Bismuth Ferrite Films” by Ming-Min Yang *et al.*, Manuscript No. NCOMMS-18-34631C.

We would like to thank the reviewers again for assessing our work. Below, we have listed the issues/comments raised by the reviewers and our corresponding replies. We believe that we addressed all the issues raised by the Reviewers, thus making the message delivered by our manuscript clearer for readers.

Response to Reviewer #3

Question 3.1 I think that the authors made the detailed reply to my previous question and included the previous discussion related to the depolarization field at the phase boundary. The reason why the conductance ratio of R/T' phase present the sinusoidal behavior is that the amplitude of the T'/T photocurrent ratio is much smaller than that of R/T ratio. To make the paper more solid, I suggest authors to change the figure 4 and/or clearly mention the reason for the sinusoidal behavior in the conductance ratio of R/T' phase. I will prevent any confusion of the readers. However, such change can be easily addressed. I have no further objection for the publication of this paper in Nature Communication.

Answer 3.1 We appreciate Reviewer's positive comment on our work and the recommendation for publication. Per the Reviewer's suggestion, we have clearly mentioned the reason for the sinusoidal behavior in the conductance ratio of R/T' phase in the manuscript.

Action Taken 3.1 Following sentence has been added to the manuscript marked as red:

“Due to the small variation amplitude of the photoconduction ratio between T' and T phase, the photoconduction ratio between R -phase and the T' -phase can also be well fitted by Eq. 1 (see Fig. 4c).”